# Macrophage internalization creates a multidrug-tolerant fungal persister reservoir and facilitates the emergence of drug resistance

Amir Arastehfar[1,5], Farnaz Daneshnia[1,2,5], Nathaly Cabrera[1], Suyapa Penalva-Lopez[1], Jansy Sarathy[1,3], Matthew Zimmerman[1,3], Erika Shor [1,3] ✉ & David S. Perlin [1,3,4] ✉

*Candida glabrata* is a major fungal pathogen notable for causing recalcitrant infections, rapid emergence of drug-resistant strains, and its ability to survive and proliferate within macrophages. Resembling bacterial persisters, a subset of genetically drug-susceptible *C. glabrata* cells can survive lethal exposure to the fungicidal echinocandin drugs. Herein, we show that macrophage internalization induces cidal drug tolerance in *C. glabrata*, expanding the persister reservoir from which echinocandin-resistant mutants emerge. We show that this drug tolerance is associated with non-proliferation and is triggered by macrophage-induced oxidative stress, and that deletion of genes involved in reactive oxygen species detoxification significantly increases the emergence of echinocandin-resistant mutants. Finally, we show that the fungicidal drug amphotericin B can kill intracellular *C. glabrata* echinocandin persisters, reducing emergence of resistance. Our study supports the hypothesis that intra-macrophage *C. glabrata* is a reservoir of recalcitrant/drug-resistant infections, and that drug alternating strategies can be developed to eliminate this reservoir.

The yeast *Candida glabrata* is a component of the human microbiome that inhabits a wide range of mucosal surfaces and is a prevalent opportunistic fungal pathogen causing bloodstream infections around the world[1–5]. A prominent increase in azole resistant *Candida* species, including *C. glabrata*, in recent years has promoted the use fungicidal (cidal) echinocandin drugs, such as caspofungin and micafungin, which interfere with cell wall biosynthesis, as prophylaxis and first line therapy against candidemia[6,7]. In turn, this change in practice has been accompanied by high rates of echinocandin- and multidrug resistant *C. glabrata* isolates in numerous clinical centers[6]. This worrying spike in

the number of drug resistant *C. glabrata* isolates, combined with the cytotoxicity of polyenes, such as amphotericin B, has greatly limited therapeutic options to treat candidemia[8].

Numerous studies have documented that echinocandin-resistant (ECR) *C. glabrata* isolates emerge from genetically related susceptible cells during the course of infection[9–14]. It has been hypothesized that a subpopulation of echinocandin-susceptible *C. glabrata* cells survives fungicidal concentrations of echinocandins in the host and this leads to the emergence of ECR isolates. In bacteriology, such cells, termed "persisters", have been well-studied[15,16]. In planktonic cultures,

[1]Center for Discovery and Innovation, Hackensack Meridian Health, Nutley, NJ 07110, USA. [2]Institute of Biodiversity and Ecosystem Dynamics (IBED), University of Amsterdam, Amsterdam 1012 WX, The Netherlands. [3]Department of Medical Sciences, Hackensack Meridian School of Medicine, Nutley, NJ, USA. [4]Georgetown University Lombardi Comprehensive Cancer Center, Washington, DC 20057, USA. [5]These authors contributed equally: Amir Arastehfar, Farnaz Daneshnia. ✉e-mail: erika.shor@hmh-cdi.org; david.perlin@hmh-cdi.org

bacterial persisters are defined by their biphasic killing dynamic, where a growth-restricted subpopulation can tolerate and survive high concentrations of cidal antibiotics lethal to their clonal susceptible kin. Persisters can be induced by environmental stresses, such as starvation, low pH, and reactive oxygen species (ROS)[15–23]. Thus, being internalized by host cells and exposed to host cell-induced stresses increases antibiotic tolerance in bacteria such as *Mycobacetrium tuberculosis, Listeria monocytogenes,* and *Staphylococcus aureus* compared to counterparts grown in culture[15–23]. Unlike antibiotic-resistant cells, persisters do not carry heritable mutations, and their progeny retain antibiotic susceptibility. After the cessation of antibiotic treatment, such persisters can reinitiate growth, causing a relapse of infection. Importantly, persisters can acquire the genetic mutations leading to mechanism-specific antibiotic resistance[15,16].

In mycology, it is assumed that persisters are formed predominantly in biofilms following exposure to fungicidal drugs, but rarely under planktonic conditions[24]. Interestingly, it has also been suggested that *C. glabrata* isolates poorly produce persister cells even in the context of biofilms[24–26]. Our group, however, has characterized a small subpopulation of *C. glabrata* persisters in vitro (which we referred to as "drug-tolerant cells") surviving supra-high MIC concentration of echinocandins[27,28]. Such subpopulations growing or surviving supra-high MIC concentrations of static or cidal antifungal drugs, respectively, have been observed in multiple fungal species and proposed to be involved in therapeutic failure, emergence of resistance, and poor clinical outcomes[29–32]. As described above, bacterial persisters are induced by environmental stress and internalization by host immune cells. It has been shown that *C. glabrata* can survive and replicate inside macrophages[1]. However, how intracellular *C. glabrata* (ICG) responds to antifungal drugs, particularly the cidal echinocandins, has not been extensively studied. Although one report showed that ICG may be killed less effectively by micafungin than *C. glabrata* growing in culture medium[33], it has not been examined whether macrophages may represent a persister reservoir in the context of infection.

Here, we show that internalization by macrophages significantly increases the abundance of multidrug-tolerant *C. glabrata* persister cells resulting in a significantly higher rate of ECR colonies. Interestingly, isolates lacking genes involved in ROS detoxification had greatly increased formation of ECR mutants, pointing to ROS as a potential inducer of mutagenesis. Finally, amphotericin B, a fungicidal drug with a different mode of action, was able to kill *C. glabrata* persisters induced by micafungin and significantly reduce ECR colony formation. Altogether, this study demonstrates that *C. glabrata* harbored within host macrophages during infection constitutes a persister reservoir and a source of ECR isolates, and points towards strategies to eliminate persisters and reduce emergence of drug-resistant strains.

## Results

### ICG shows a persister phenotype
To assess the impact of antifungal drugs on ICG, we selected two isolates with similar MIC values (Supplementary Table 1): reference strain CBS138 (sequence type 15, ST15) and strain Q36, a clinical isolate belonging to ST3. THP1 macrophages exposed to *C. glabrata* cells for 3 h were extensively washed to remove non-adherent yeast cells and treated with RPMI containing 2X MIC of the fungistatic azoles fluconazole and voriconazole, and fungicidal echinocandins micafungin, and polyene amphotericin B. The two strains were also incubated in RPMI medium alone containing the same drug concentrations. The survival/proliferation of both ICG and planktonic cells were measured by colony forming unit (CFU) counts and normalized against respective untreated controls at 3, 6, and 24 h post treatment (pst). Interestingly, drug-treated ICG were significantly less sensitive to all drugs relative to planktonic cells, suggesting that the fungistatic azoles did

not inhibit ICG proliferation whereas the fungicidal echinocandins did not kill the ICG as effectively as planktonic cells (Fig. 1a). Next, we treated ICG and planktonic cells with a wide range of micafungin concentrations (2X-256X MIC) and observed that the survival of ICG was 100–1000-fold higher than that of planktonic cells, especially at 24 h pst (Fig. 1b). This high survival rate of ICG was reminiscent of intracellular bacterial persister cells[15,16]. To ask whether ICG fulfill other criteria for being considered persisters, we selected a single micafungin concentration (0.125 μg/ml, 8X MIC) and investigated multiple parameters used to specify bacterial persisters.

First, to ask whether ICG show the hallmark persister biphasic killing, THP1 macrophages were infected with *C. glabrata* and incubated for 3 h. Subsequently, non-adherent yeast cells were removed by extensive washing and survival was assessed at 1-, 3-, 6-, and 24-h pst. Indeed, the ICG showed the typical biphasic killing to both micafungin and amphotericin B (Fig. 1c).

To test whether ICG cells are non-proliferating, prior to infecting macrophages we stained the initial inoculum with isothiocyanate (FITC), which does not transfer to daughter cells. At designated timepoints, the ICG was counterstained with Alexa Flour-647 (AF647), which stains all cells (mothers and daughters), and the micafungin-treated ICG and untreated controls were released from macrophages and subjected to flow cytometry. *C. glabrata* cells single positive for AF647 represented daughter cells and therefore indicated proliferation within macrophages, whereas cells doubly stained with both FITC and AF647 represented the mother cells. Interestingly, while untreated ICG showed a dramatic expansion of daughter cells, upon micafungin treatment the proportion of mother and daughter cells did not vary at 3, 6, 24, or 48 h of micafungin exposure (Fig. 1d), indicating low or no proliferation.

Next, we tested whether after a 24-h exposure to micafungin ICG had acquired genetic determinants of echinocandin resistance (mutations in the hot-spot (HS) regions of *FKS1* and *FKS2*). Treated ICG was plated on YPD containing 0.125 μg/ml micafungin (8xMIC) to capture the ECR colonies. We selected this concentration because our pilot studies had shown that it could detect various *FKS* HS mutations after plating a range of cell numbers (10–10^6 cells). We found that ICG did not produce any ECR colonies, consistent with a lack of *fks* mutations in ICG, and that colonies obtained from them showed killing dynamics similar to the parental strain.

Finally, we asked whether *C. glabrata* released from micafungin-treated macrophages are culturable and whether they are capable of initiating another cycle of infection and proliferation inside macrophages. After a 24-h micafungin treatment, *C. glabrata* cells were released from macrophages and stained with propidium iodide (PI). PI-negative (PI−) cells were selected and either plated on YPD plates or used to infect macrophages, and proliferation within macrophages was assessed after 72 h. We found that *C. glabrata* released from micafungin-treated macrophages were capable of forming colonies (Fig. 1e) and reinfecting and proliferating within macrophages (Fig. 1f), but at significantly lower rates than planktonic cells, which is similar to results obtained with bacterial persisters[34].

### Micafungin efficiently penetrates into macrophages and rapidly reaches supra-MIC concentrations
We sought to determine the dynamics of micafungin penetration into the macrophages to assess if the high survival of ICG cells was simply due to poor micafungin penetration. We directly measured the concentration of micafungin inside THP1 macrophages cultured in 4 μg/ml (3.15 μM) of the drug using liquid chromatography with tandem mass spectrometry (LC/MS/MS), which can accurately predict drug penetration at the site of infection in vivo[35,36]. We measured the intracellular (IC) and extracellular (EC) concentrations of micafungin at 30 min and 3 and 24 h pst. We included three control drugs with well-established macrophage penetration capacities, each set at

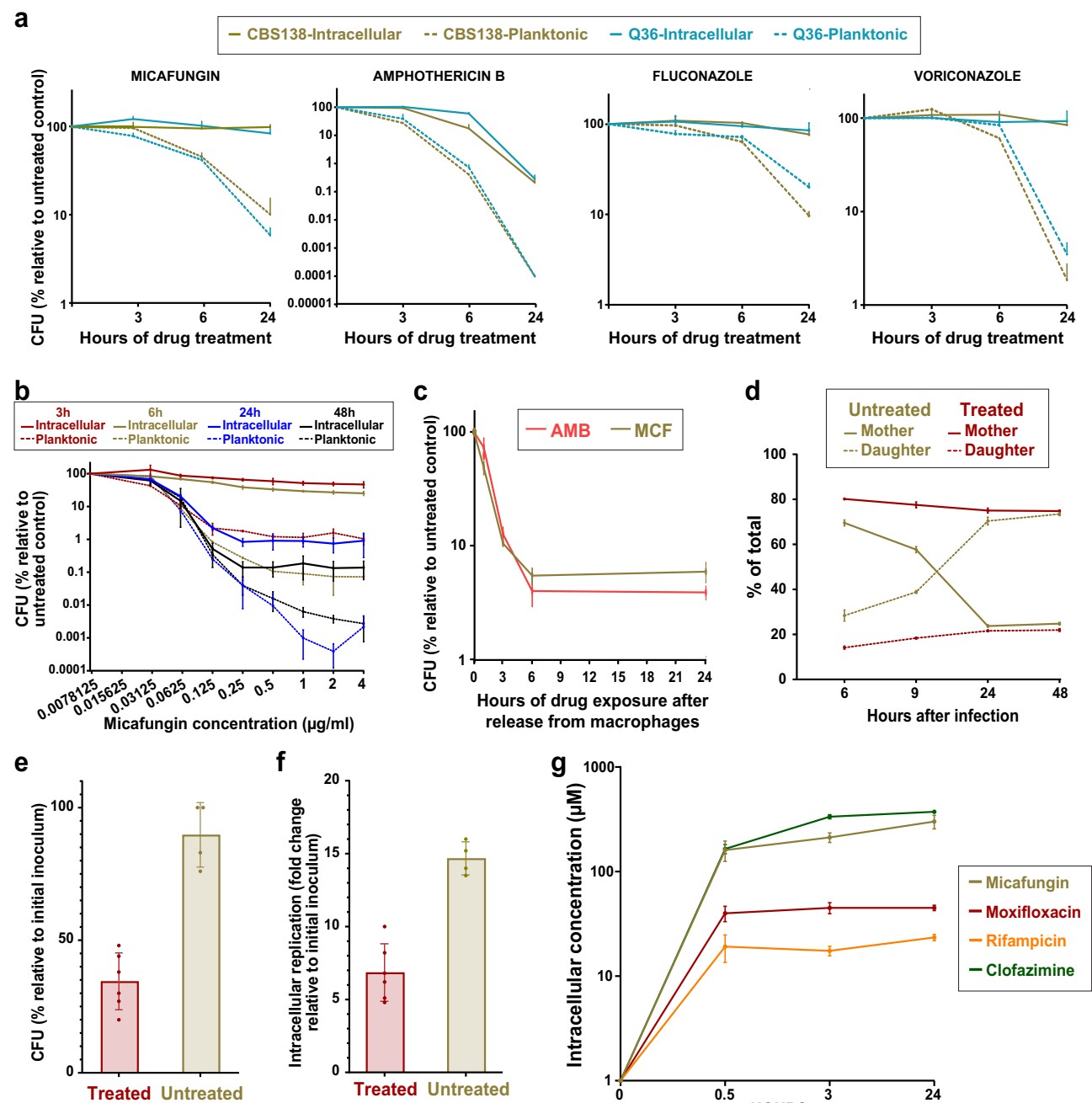

**Fig. 1 | Intracellular *C. glabrata* (ICG) cells show multiple characteristics of persisters. a** ICG are less sensitive to micafungin (0.03 μg/ml), AMB (1 μg/ml), fluconazole (8 μg/ml), and voriconazole (0.06 μg/ml) than their planktonic counterparts. *n* = 4 biological replicates over two independent experiments. **b** ICG survival was higher than that of planktonic cells over a range of micafungin concentrations. *n* = 4 biological replicates over two independent experiments. **c** ICG exposed to 8X MIC of micafungin (0.125 μg/ml) or AMB (4 μg/ml) show a biphasic killing pattern. *n* = 3 biological replicates over 3 independent experiments. **d** Proportions of daughter (AF647 ConA-positive) and mother cells (FITC- and AF-647 ConA-positive) in micafungin-treated ICG remain stable over the course of treatment, indicating a lack of proliferation, whereas untreated ICG show a marked increase in the proportion of daughter cells. *n* = 3 biological replicates in a single experiment. **e** Propidium iodide (PI)-negative *C. glabrata* cells released from micafungin-treated macrophages have lower culturability, as assessed by colony forming units (CFU), than PI-negative *C. glabrata* released from untreated macrophages. *n* = 4–6 biological replicates over two independent experiments. **f** PI-negative *C. glabrata* released from micafungin-treated macrophages are capable of reinfecting new macrophages, albeit at a lower rate relative to untreated counterparts. *n* = 4–5 biological replicates over two independent experiments. **g** Micafungin penetrated macrophages and accumulated there at a high concentration, similar to clofazimine and higher than moxifloxacin and rifampicin. *n* = 3 biological replicates in a single experiment. PI propidium iodide, AF-647 ConA Alexa Flour 647-conjugated con-canavalin A, FITC fluorescein isothiocyanate, ICG intracellular *C. glabrata*, MCF micafungin AMB amphotericin B, MIC minimum inhibitory concentration, CFU colony forming units, LC-MS/MS liquid chromatography-tandem mass spectro-metry. Data are presented as mean values +/− standard deviations.

5 μM: rifampicin (poorly penetrating), moxifloxacin (moderately penetrating), and clofazimine (highly penetrating). As expected and consistent with previous studies[36–38], rifampicin showed the poorest penetration, whereas moxifloxacin and clofazimine moderately and highly penetrated the macrophages, respectively (Fig. 1g). Micafungin showed comparable penetration into the macrophages to clofazimine despite being used at a lower concentration. This result was consistent with previously reported strong penetration and accumulation of echinocandins in macrophages in clinical settings[39–41] and indicated that ICG retain high viability even though the intracellular micafungin concentration reaches >100 μM, or 256X of the MIC, by 0.5 h pst (Fig. 1g).

### Internalization of *C. glabrata* cells by THP1 macrophages significantly increases their tolerance to cidal antifungals

Various environmental stresses are well known to induce cross-stress tolerance, including tolerance to antibiotics and antifungals. In particular, macrophage internalization increases the number of antibiotic persisters by exposing bacteria to an array of stresses[15–23]. Accordingly, we tested whether internalization of *C. glabrata* by macrophages increased their cidal drug tolerance. *C. glabrata* cells were used to infect macrophages, cultured for 3, 6, 24, and 48 h, then released from macrophages, exposed to micafungin (0.06 μg/ml or 4xMIC) for 1 h, and plated on drug-free YPD for CFU counts (Fig. 2a). Control planktonic cells were cultured in RPMI but otherwise treated identically. Interestingly, ICG showed a significantly higher survival after treatment, especially at 3 h, compared to planktonic cells (Fig. 2b). Also, consistent with the observation that stationary phase bacteria are significantly enriched for persisters[42], we found that planktonic *C. glabrata* cells at 24 and especially 48 h showed the highest survival rates upon micafungin exposure (Fig. 2b).

To assess whether internalization by macrophages renders ICG more tolerant to other cidal antifungals, *C. glabrata* cells were used to infect macrophages or inoculate RPMI and 3 h later exposed to 4X MIC of caspofungin or amphotericin B for 1 h. Interestingly, ICG again showed a significantly higher survival in both caspofungin and amphotericin B relative to their planktonic counterparts (Fig. 2c).

Finally, to understand how long the higher drug tolerance of ICG cells lasts, exponentially growing *C. glabrata* cells (CBS138) were either incubated in RPMI or internalized by macrophages for 3 h, released, and then exposed to either micafungin (0.125 μg/ml) or amphotericin B (4 μg/ml) for 1, 3, 6, or 24 h. Strikingly, the high tolerance level of ICG cells was detected up until 24 h and was the highest at 24-h time-point, especially for micafungin (Fig. 2d). Altogether, these experiments showed that ICG cells display a long-lasting multidrug-tolerant phenotype.

### *C. glabrata* cells harbored in the spleen have higher caspofungin tolerance than those harbored in the kidney

Because ICG showed significantly higher survival after release from macrophages, we asked whether during systemic infection *C. glabrata* harbored in the spleen, which contains one of the largest macrophage populations in the body[43], may have higher echinocandin tolerance than *C. glabrata* cells harbored in another organ. Echinocandin efficacy in this model is evaluated using fungal burdens in the organs, as it does not result in lethality. Mice were systemically infected with CBS138 and either treated with a humanized dose of caspofungin (5 mg/kg) daily starting at 24 h post-infection or received PBS alone (control). Spleen and kidney were harvested after 2 and 6 days of treatment, homogenized, and plated on YPD for CFU counts. We found that in vehicle-treated mice, spleen fungal burdens decreased more than kidney fungal burdens, perhaps due to the activity of spleen macrophages (Supplementary Fig. 1a). Interestingly, whereas kidney *C. glabrata* burdens significantly decreased in caspofungin-treated mice, spleen fungal burdens were almost unaffected by caspofungin treatment (Fig. 2e, Supplementary Fig. 1a), indicating that macrophage-rich anatomical niches likely contain drug-tolerant persister *C. glabrata* cells. To determine whether this result could be due to insufficient caspofungin penetration into the spleen, caspofungin concentrations were measured in the same organs at the same time-points. We found that caspofungin effectively penetrated both organs, and that, although caspofungin concentration in the spleen was about 2-fold lower than that in the kidney, both concentrations greatly exceeded the strain's caspofungin MIC (0.25 μg/ml) (Supplementary Fig. 1b). For instance, by day 6 of treatment the spleen caspofungin concentration exceeded the MIC by 48-fold, yet there was no evidence of the drug's efficacy in this organ. Together, these data support the conclusion that caspofungin is less active against *C. glabrata* in the spleen than in the kidney, which is likely due to the high abundance of intra-macrophage *C. glabrata* in the former but may also be influenced by somewhat lower drug penetration.

### ROS produced by THP1 macrophages renders *C. glabrata* more tolerant to micafungin

To understand which stress applied by macrophages renders *C. glabrata* tolerant to cidal antifungal drugs, we carried out an in vitro experiment, in which *C. glabrata* cells were incubated in RPMI containing 10 mM $H_2O_2$ (representing ROS), pH5 RPMI (representing vacuolar acidity), spent RPMI (representing nutrient-poor vacuolar environment), spent RPMI + 10 mM $H_2O_2$, or regular control RPMI. Next, at designated time-points the *C. glabrata* cells were exposed to micafungin (0.06 μg/ml or 4xMIC) for 1 h. Interestingly, *C. glabrata* cells incubated in either spent RPMI + $H_2O_2$ or $H_2O_2$ alone (and, to a lesser extent, those in spent RPMI) showed significantly increased tolerance to micafungin, whereas those incubated in acidic medium showed similar killing rates as controls (Fig. 3a).

To determine how soon the combination of nutrient deprivation and ROS induces drug tolerance in *C. glabrata*, we incubated *C. glabrata* cells in spent RPMI + $H_2O_2$ for 5, 15, 30, 60, and 120 min, exposed the cells to 0.06 μg/ml micafungin (4xMIC) for 1 h, and assessed survival by CFU counts. Consistent with studies of bacterial persisters[19,21,23,44], we found that a high level of tolerance can be induced as early as after 15 min of incubation in spent RPMI + $H_2O_2$ (Fig. 3b and Supplementary Fig. 1).

To test directly whether macrophage-produced ROS enhances ICG drug tolerance, THP1 macrophages were pretreated with an antioxidant (butylated hydroxanisole, BHA) or a NAPH oxidase inhibitor (VAS2870) prior to *C. glabrata* infection[21]. We also pretreated macrophages with bafilomycin A (Baf A), a vacuolar ATPase inhibitor that prevents phagosome acidification[45]. Untreated macrophages served as controls. We found that ICG released from macrophages pretreated with BHA or VAS2870 showed significantly lower survival than ICG released from untreated macrophages (Fig. 3c), indicating that macrophage ROS are a key inducer of ICG drug tolerance. Of note, ICG released from BafA-treated macrophages also had a lower survival rate than controls, which may indicate vacuole acidification may also partly contribute to the higher tolerance of ICG to micafungin.

It has been shown that *C. albicans* and *C. glabrata* cells internalized by macrophages for 3 h suppress their transcription and translation machineries[46,47]. Thus, we tested whether inhibiting transcription or translation using sublethal concentrations of thiolutin and cycloheximide[1], respectively, could enhance the tolerance of *C. glabrata* cells to micafungin. Indeed, inhibition of either transcription or translation significantly increased *C. glabrata* micafungin tolerance (Fig. 3d). Together, these data indicate that macrophage engulfment exposes *C. glabrata* to environmental stresses (ROS, nutrient deprivation, and low pH), prompting it to downregulate transcription and translation and rapidly acquire a drug-tolerant state, leading to persistence.

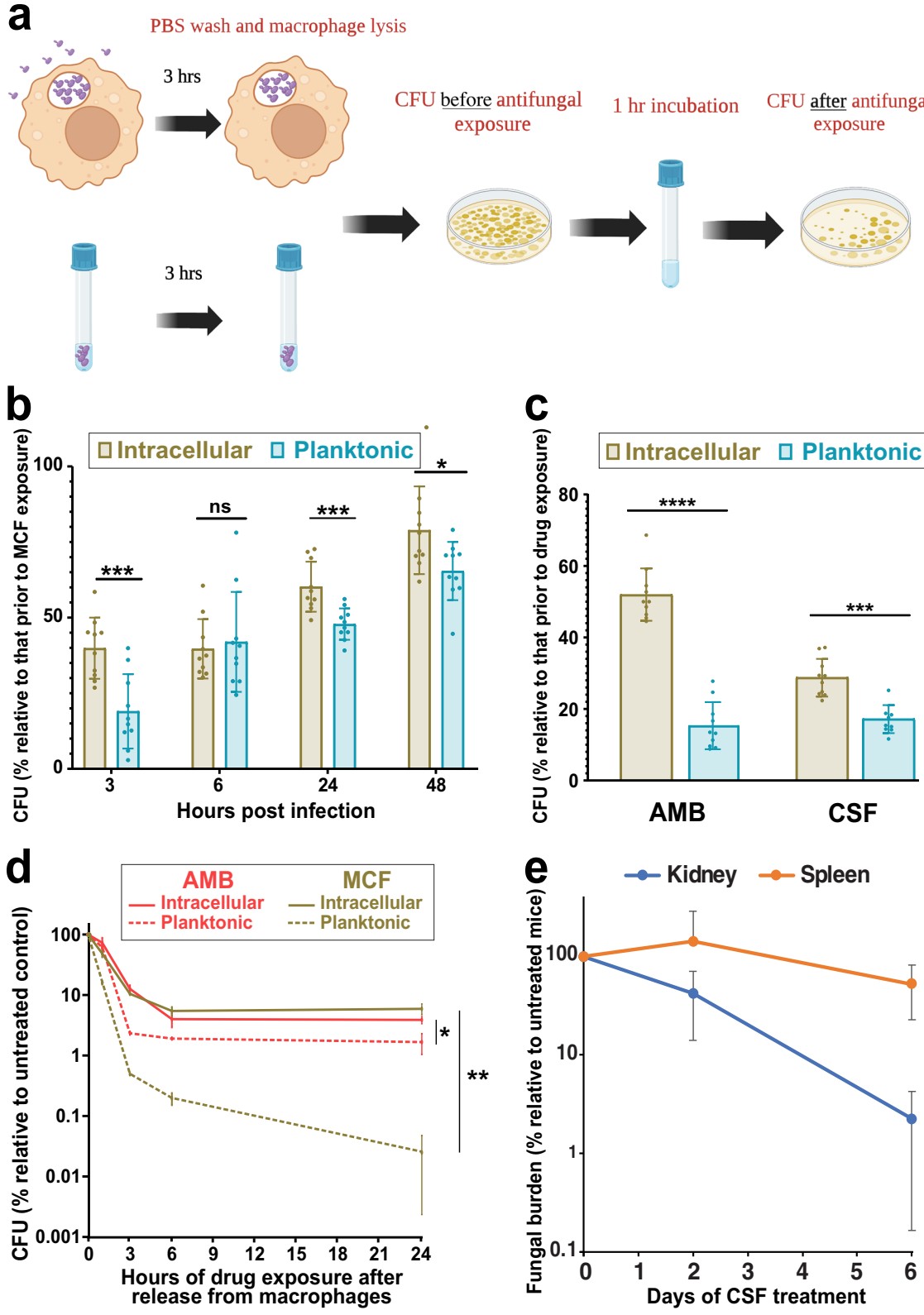

## Internalization by macrophages significantly increases the emergence of ECR colonies

It has been shown that bacterial persisters constitute a reservoir from which drug-resistant colonies can emerge[48], and the same has been hypothesized for fungal cells[27,28]. Therefore, we asked whether ICG is a source of ECR mutations. We infected macrophages with 5 clinical strains belonging to different STs as well as the reference strain CBS138

and monitored the emergence of ECR mutations in ICG in the presence of micafungin (0.125 μg/ml or 8xMIC). To detect ECR cells, *C. glabrata* were released from macrophages at 24-h intervals and plated on micafungin-containing YPD plates (0.125 μg/ml) (Fig. 4a). The initial inoculum of each isolate was also plated on YPD + micafungin to verify that it did not contain ECR cells. Of note, the majority of clinical ECR isolates associated with therapeutic failure harbor mutations in HS1 of

**Fig. 2 | Internalization of *C. glabrata* cells by macrophages enhances their tolerance to cidal antifungal drugs. a** *C. glabrata* cells either incubated in RPMI or macrophages for 3 h and their survival were assessed before and after 1 h exposure to micafungin (0.06 μg/ml). **b** *C. glabrata* released from macrophages at different times post-infection showed significantly higher survival upon treatment with 0.125 μg/ml micafungin for 1 h compared to similarly treated planktonic cells. $n = 10$ biological replicates over two independent experiments (*$p < 0.03$, **$p < 0.003$, ***$p < 0.0003$, two-sided paired t-test). **c** *C. glabrata* released from macrophages 3 h post-infection and treated for 1 h with either caspofungin (0.125 μg/ml) or amphotericin B (2 μg/ml) showed higher survival than similarly treated planktonic cells. $n = 10$ biological replicates over two independent experiments.

(***$p < 0.0003$, ****$p < 0.00003$, two-sided paired t-test). **d** *C. glabrata* cells released from macrophages 3 h post-infection and treated with 0.125 μg/ml micafungin or 2 μg/ml amphotericin B showed significantly higher survival than similarly treated planktonic cells over a wide range of time-points. $n = 3$ biological replicates over two independent experiments (*$p < 0.03$, **$p < 0.003$, two-sided paired t-test). **e** *C. glabrata* burden in the spleen (a macrophage-rich organ) was not significantly reduced by treatment with a humanized dose of caspofungin (5 mg/kg), in contrast to the kidney. $n = 5$ biological replicates (mice) in one experiment. MCF micafungin, AMB amphotericin B, CSF caspofungin, CFU colony forming units. Data are presented as mean values +/− standard deviations.

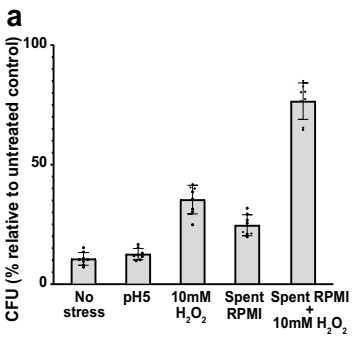
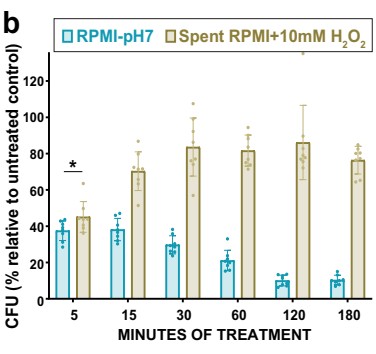
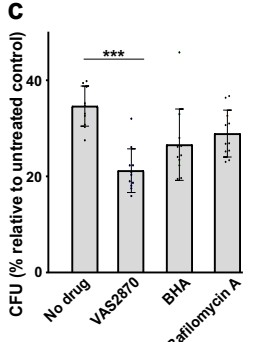
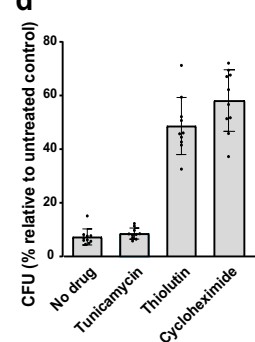

**Fig. 3 | Oxidative stress and starvation drive *C. glabrata* into a drug-tolerant, persister-like state. a** Oxidative stress (RPMI + 10 mM $H_2O_2$), starvation (spent RPMI), and especially oxidative stress in combination with starvation (spent RPMI + 10 mM $H_2O_2$) increased the survival of *C. glabrata* after exposure to 0.125 μg/ml micafungin, whereas low pH (RPMI pH5) had no effect. $n = 8$ biological replicates over two independent experiments. **b** The effect of oxidative and starvation stresses on survival was evident as early as after 15 min of stress. $n = 8$ biological replicates over two independent experiments. (*p < 0.03, modified two-sided t-test). **c** Inhibiting macrophage ROS production by using a NOX inhibitor (VAS2870) or an antioxidant (BHA) resulted in decreased survival of macrophage-released *C. glabrata* after 1 h in 0.06 μg/ml micafungin, whereas treating macrophages with a vacuolar ATPase inhibitor (Bafilomycin A) had a weaker effect. $n = 11$ biological replicates over 3 independent experiments. **d** *C. glabrata* cells incubated in RPMI containing sublethal concentrations of a transcription inhibitor (thiolutin 5 μg/ml) or a translation inhibitor (cycloheximide 1 μg/ml) had a significantly higher tolerance to micafungin (0.06 μg/ml) compared to *C. glabrata* treated with an endoplasmic reticulum stressor (tunicamycin 5 μg/ml) or untreated cells. $n = 11$ biological replicates over 3 independent experiments. ICG intracellular *C. glabrata*, NOX NADPH oxidase, BHA butylated droxyanisole, ROS reactive oxygen species, RPMI Roswell Park Memorial Institute. Data are presented as mean values +/− standard deviations.

*FKS2*, followed by HS1 of *FKS1*[49,50]. Thus, the ECR colonies growing on YPD + micafungin plates were subjected to sequencing of the four *FKS* HSs (*FKS1* HS1 and 2, *FKS2* HS1 and 2)[51]. As in previous experiments (Fig. 1a and Fig. 3c), both ICG and planktonic *C. glabrata* showed declining survival over time up to 48 h of micafungin treatment, with ICG being significantly more tolerant to micafungin than planktonic cells (Fig. 4b). After 48 h, however, micafungin-treated planktonic cells displayed a CFU rebound (Fig. 4b), which is reminiscent of phenotypic resistance[16]. The basis for phenotypic resistance is not clear, although it may result from decreased effective drug concentrations due to drug binding to fragments (e.g., pieces of membrane and cell wall) of dead cells, as has been hypothesized before for bacteria[52]. Nevertheless, ICG produced a significantly higher frequency of ECR colonies compared to planktonic cells, and all these ECR colonies harbored well-known and clinically relevant mutations in HS1 of *FKS2* (Fig. 4b, Supplementary Table 1). This result indicates that macrophage-engulfed *C. glabrata* may constitute a reservoir from which ECR *C. glabrata* cells can emerge.

### Deletion of ROS detoxifying genes markedly increases the frequency of ECR emergence

ROS are generally thought to be detrimental to cells because of their capacity to damage cellular macromolecules. However, both bacterial and fungal cells also use endogenously generated ROS to induce programmed genetic instability under specific circumstances[53,54]. Our group previously showed that although echinocandins induce endogenous ROS in *C. glabrata*, the use of ROS scavengers did not enhance the survival of *C. glabrata* in echinocandin presence and that, surprisingly, *C. glabrata* downregulated ROS detoxifying genes upon echinocandin exposure[27]. To ask whether ROS are involved in persistence and emergence of ECR mutants, we deleted genes with well-known functions in ROS detoxification: catalase (*CAT1*, CAGL0K10868g), glutathione oxidoreductase (*GRX2*, CAGL0K05813g), manganese superoxide dismutase (*SOD2*, CAGL0E04356g), and three transcription regulators of oxidative stress responses, namely *SKN7* (CAGL0F09097g), *MSN4* (CAGL0M13189g), and *YAP1* (CAGL0H04631g)[27,55]. In addition, because it has been shown that in *Mycobacetrium tuberculosis* isocitrate lyase mutants have elevated antibiotic-induced ROS levels[56], we also deleted *C. glabrata* isocitrate lyase gene *ICL1* (CAGL0L09273g).

All deletion mutants grew normally in YPD broth (Fig. 5a). Consistent with their compromised ROS detoxification activities, most of the mutants showed increased ROS levels in the presence of micafungin (0.125 μg/ml or 8xMIC) (Fig. 5b). To measure their sensitivity to exogenous ROS, we measured their survival in RPMI containing 10 mM of $H_2O_2$. Consistent with a previous study[53], we found that deleting *CTA1* and the two main transcription factors regulating its expression, *YAP1* and *SKN7*, significantly impaired strain survival in the presence of $H_2O_2$, whereas the other mutants were unaffected (Fig. 5c). We also measured the mutants' survival and proliferation within macrophages and found that *skn7Δ*, *sod2Δ*, and *yap1Δ* mutants showed significantly reduced survival 24 h after infection, whereas only the *sod2Δ* mutant showed significantly reduced survival 48 h after infection (*cta1Δ* and *icl1Δ* mutants did not reach the statistical significance threshold with

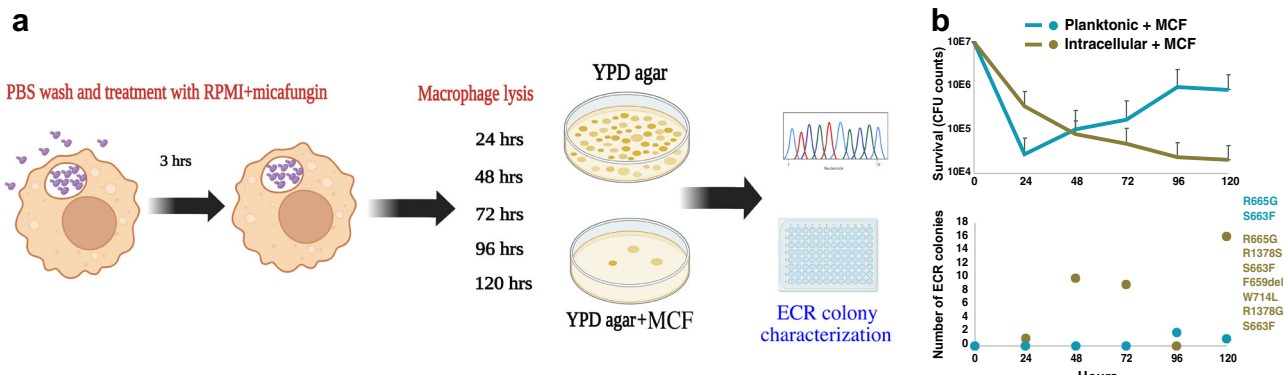

**Fig. 4 | ICG develops echinocandin-resistant mutations. a** Echinocandin resistance and survival rates of *C. glabrata* cells incubated in macrophages and RPMI containing micafungin (0.125 μg/ml) were evaluated. At each time-point, *C. glabrata* cells were collected, a portion was spread on YPD to assess survival and the other portion was plated on YPD plates containing micafungin (0.125 μg/ml) to determine the ECR rate. **b** The dynamics of killing (top panel) and the numbers of ECR colonies (bottom panel) detected at different time-points are shown for both ICG and planktonic cells cultured in the presence of 0.125 μg/ml micafungin. *n* = 3 biological replicates over 3 independent experiments. ICG intracellular *C. glabrata*, ECR echinocandin resistant. Data are presented as mean values +/− standard deviations.

*p* = 0.06 for both) (Fig. 5d). These results suggested that either the *C. glabrata* ROS detoxification pathways are redundant[57] or the ROS levels inside macrophages were not high enough to strongly impair the mutants.

Next, we assessed the mutants' survival in the presence of micafungin and the emergence of ECR mutations either when cultured in RPMI or harbored within macrophages. Interestingly, although none of the mutants were significantly impaired for survival in the presence of micafungin, both ICG and planktonic ROS detoxification mutants (except *icl1Δ*) showed greatly increased ECR colony emergence relative to the WT strain (Fig. 5e and Supplementary Table 2). ECR frequencies were even higher in planktonic cells than in ICG, suggesting that it is not macrophage-produced exogenous ROS but *C. glabrata*-produced endogenous ROS may promote this mutagenesis. Interestingly, we did not recover any ECR colonies from the *icl1Δ* strain, although it had no obvious survival defect, despite repeating the experiment 3 times with a 120 h follow-up. This observation may point to the potential importance of *ICL1* for cellular processes linked to mutagenesis. Given the absence of this gene in human genome and that this gene is evolutionarily conserved across a wide range of pathogens[58], it may be a promising target to reduce emergence of ECR mutants in *C. glabrata*.

**Micafungin-tolerant ICG are effectively killed by amphotericin B**
We have shown that the nutritional and ROS stresses imposed by macrophages on ICG drive the fungus into a non-proliferative state that is highly tolerant to echinocandins. In principle, this persister population may be susceptible to a different class of drug whose cidal activity is not strongly affected by the metabolic or proliferative state of the pathogen. Indeed, in the bacterial field cidal antibiotics have been classified as either strongly dependent on metabolism (SDM) or weakly dependent on metabolism (WDM)[59–61]. For *Candida* species, the metabolic dependency of cidal antifungal drugs has not been fully examined. Thus, we compared the metabolism dependencies of micafungin (0.125 μg/ml), caspofungin (0.25 μg/ml), and amphotericin B (2 μg/ml) against *C. glabrata* cells incubated in 100%, 20%, and 2% RPMI. Not surprisingly, micafungin and caspofungin were significantly affected by RPMI concentration, classifying them as SDM drugs, whereas the lethality of amphotericin B was much less dependent on RPMI concentration, classifying it as a WDM drug (Fig. 6a). The strong dependency of echinocandins on the presence of nutrients is consistent with their killing activity being dependent on proliferating cells, whereas the weak metabolic dependency of amphotericin B, a pore former, is consistent with its ability to kill both proliferating and non-proliferating cells.

Next, we asked whether amphotericin B could effectively kill ICG that had survived micafungin treatment. We treated ICG with micafungin (0.125 μg/ml or 8xMIC) for 24 h, followed by either replacing the medium with RPMI containing amphotericin B (2 μg/ml or 4xMIC) or fresh RPMI containing micafungin (0.125 μg/ml), and survival was measured at 24- and 48-h timepoints. Interestingly, our results showed that ICG treated with amphotericin B had a significantly lower survival rate compared to ICG treated with micafungin alone (Fig. 6b).

Finally, we asked whether the amphotericin B-mediated increase in ICG killing was accompanied by reduced emergence of ECR colonies. We selected three ROS detoxification mutants showing a high ECR frequency following micafungin exposure (*cta1Δ*, *skn7Δ*, and *yap1Δ*), treated them with micafungin (0.125 μg/ml) for 24 h to deplete drug-sensitive non-persisters, and then either replaced the medium with fresh RPMI containing amphotericin B (2 μg/ml) or fresh RPMI containing micafungin (0.125 μg/ml). ECR frequency was measured at 48- and 72-h timepoints. Like the WT strain, the mutants had significantly higher survival when treated only with micafungin than when treated with micafungin followed by amphotericin B, and as expected, the higher survival was accompanied by higher ECR frequencies (Fig. 6c), which could partly be explained by the fact that amphotericin B can indistinguishably kill both ECR and susceptible colonies. Collectively, these results indicate that alternating treatments with echinocandins and amphotericin B may be an effective strategy to eliminate intracellular *C. glabrata* persister reservoirs and decrease the frequency of emergence of ECR isolates.

## Discussion
*C. glabrata* is notable for its ability to proliferate inside macrophages[1,57]. Our results identify macrophages as an important host reservoir of *C. glabrata* persistence and help explain how this fungal pathogen is able to rapidly generate antifungal drug-resistant mutations during treatment[9–14]. We show that ICG cells exhibited hallmarks of bacterial persisters, such as increased survival in the presence of cidal drugs, a biphasic killing curve, lack of proliferation, lack of genetic mutations associated with mechanism-specific resistance, reduced culturability, and the ability to reinfect macrophages. We also identify macrophage-generated ROS as an important inducer of *C. glabrata* drug tolerance and demonstrate that deleting of ROS detoxification genes greatly induces the emergence of ECR mutations. Finally, we show that intra-macrophage persister cells formed during echinocandin treatment can be eliminated by amphotericin B. Together, these results provide insights into how fungal pathogens persist

in the host despite treatment with cidal antifungal drugs. These insights could be leveraged to develop more effective antifungal regimens, e.g., by alternating treatments with echinocandins with short treatments with amphotericin B, which would decrease the amphotericin B-associated toxicity while helping eliminate fungal persister cells and reducing the emergence of drug-resistant strains.

Our observation that ROS generated both in cellulo and in vitro increased the survival of *C. glabrata* upon exposure to a lethal concentration of micafungin is consistent with observations made using

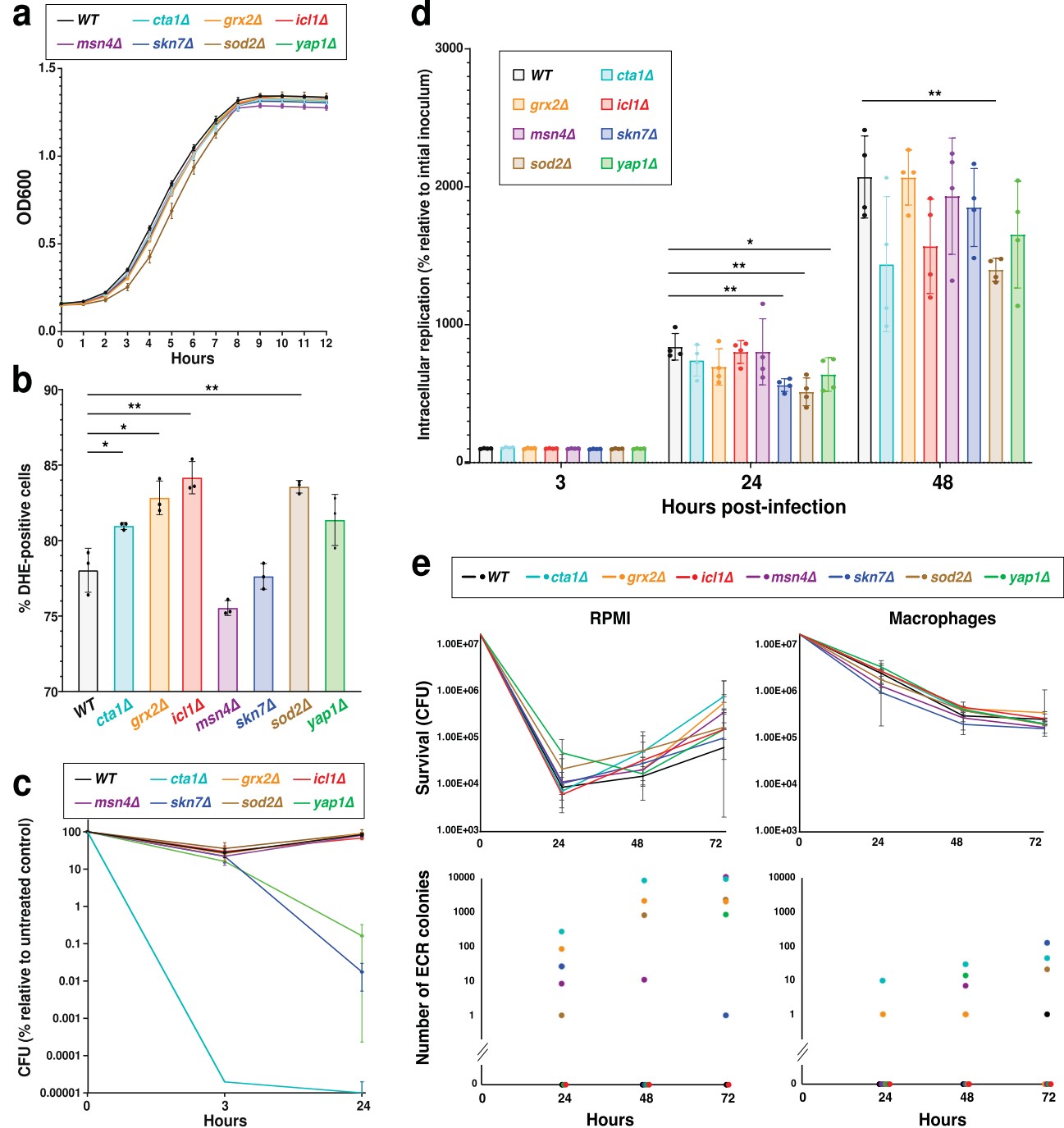

**Fig. 5 | Deletion of *C. glabrata* ROS detoxification genes promotes the emergence of ECR mutants in both ICG and planktonic cells. a** ROS mutants had normal growth rates in YPD broth. *n* = 3 biological replicates in a single experiment. **b** With the exception of *msn4Δ* and *skn7Δ*, the ROS detoxification mutants had significantly higher superoxide levels compared to the WT strain when exposed to micafungin (0.125 μg/ml), as measured by DHE straining intensity. *n* = 3 biological replicates over 3 independent experiments (*$p < 0.03$, **$p < 0.003$, two-sided paired t-test). **c** In the presence of 10 mM $H_2O_2$ *cta1Δ*, *skn7Δ*, and *yap1Δ* mutants had significantly reduced survival compared to the WT strain. *n* = 4 biological replicates over two independent experiments. **d** Several ROS mutants had mild survival/replication defects inside THP1 macrophages at 24-h and 48-h time-points. *n* = 4 biological replicates over two independent experiments (*$p < 0.03$, **$p < 0.003$, modified two-sided t-test). **e** The mutants were exposed to 0.125 μg/ml micafungin and survival and ECR colony formation were monitored. Although the mutants showed comparable survival in micafungin to the WT strain (top panel), they produced a significantly higher number of ECR colonies under either planktonic or intra-macrophage conditions (bottom panel). *n* = 4 biological replicates over 4 independent experiments. WT wild type, YPD yeast extract-peptone-dextrose, ICG intracellular *C. glabrata*, DHE dihydroethidium. Data are presented as mean values +/− standard deviations.

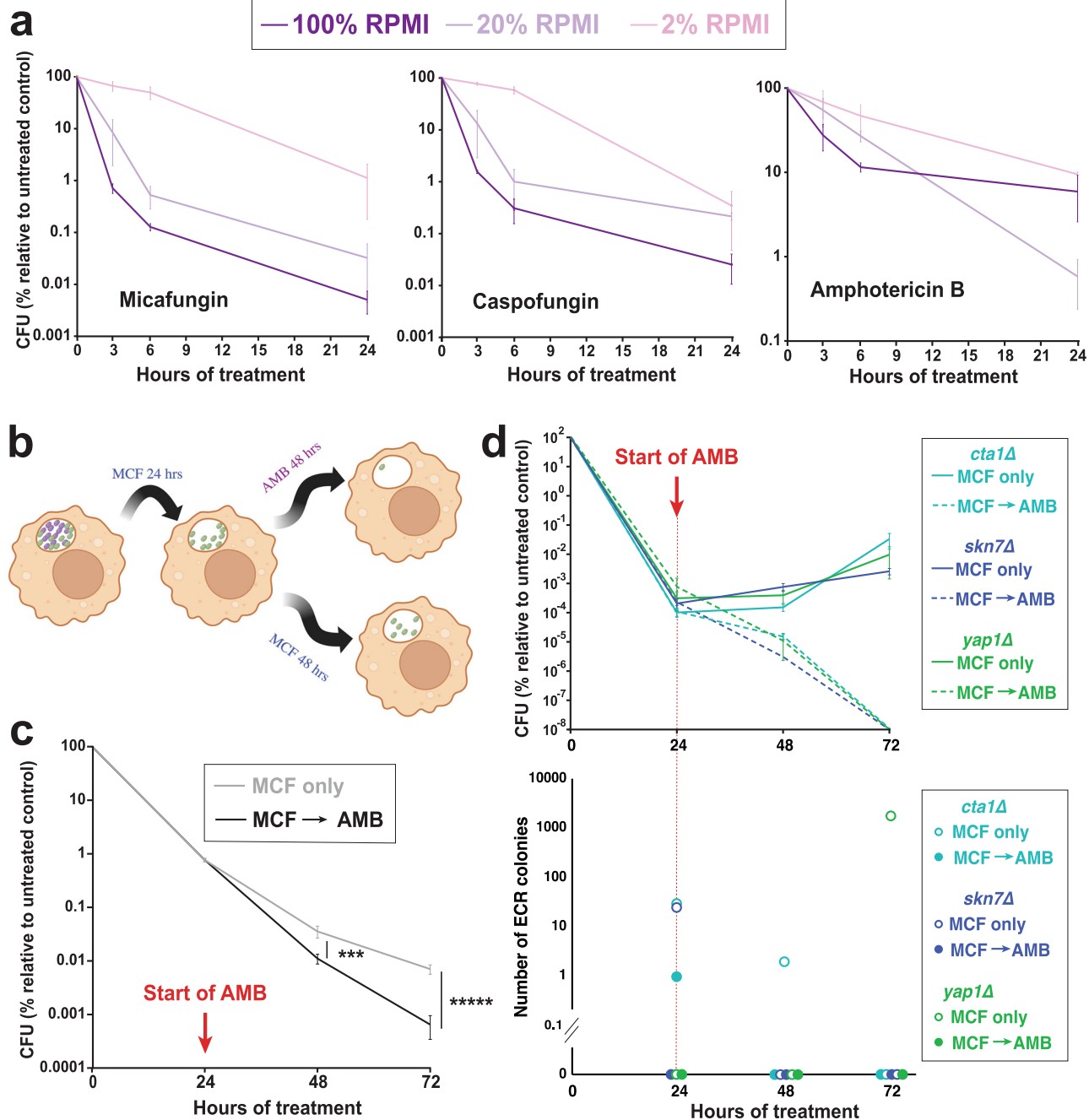

**Fig. 6 | Amphotericin B efficiently kills echinocandin persisters. a** The metabolic dependencies of micafungin, caspofungin, and amphotericin B were assessed by incubating *C. glabrata* (CBS138) at various dilutions of RPMI (100%, 20%, and 2%). **a** Micafungin and caspofungin killing efficiencies were much more affected by RPMI concentration than that of amphotericin B. $n = 6$ biological replicates over 3 independent experiments. **b** ICG were treated with micafungin (0.125 μg/ml) for 24 h, then treated with fresh RPMI containing either amphotericin B (2 μg/ml) or micafungin (0.125 μg/ml), and survival rates were monitored for the next 48 h. **c** Following micafungin with amphotericin B significantly reduced the numbers of

persister cells. $n = 7$ biological replicates over 3 independent experiments (***$p < 0.0003$, *****$p < 0.000003$, modified two-sided t-test). **d** The same experiment was repeated with planktonically growing *cta1Δ*, *skn7Δ*, and *yap1Δ* mutants, and both survival and ECR colony frequency were assessed. Both survival and ECR frequencies were significantly reduced by following micafungin treatment with amphotericin B. $n = 4$ biological replicates over 4 independent experiments. ICG intracellular *C. glabrata*, ECR echinocandin-resistant, MCF micafungin, AMB amphotericin B. Data are presented as mean values +/− standard deviations.

intracellular *Staphylococcus aureus* cells, where ROS likewise induced antibiotic tolerance[21,22]. How ROS induces drug tolerance and persistence is not well understood, but it has been shown that ROS can corrupt iron-sulfur-containing enzymes involved in tricarboxylic acid (TCA) cycle[21,22], which can result in lower respiration and ATP depletion[44,62]. Echinocandins' killing activity may be more dependent on the metabolic/proliferative state of fungal cells because their target—the beta-glucan synthase enzyme—is specifically required by growing cells building and remodeling their cell walls. In contrast, amphotericin B can form pores in the membranes of both growing and non-growing cells, helping explain its weak dependence on the fungal metabolic status.

Interestingly, our previous RNAseq analysis of *C. glabrata* treated with echinocandins in vitro showed a strong increase in ROS levels but a concomitant downregulation of genes involved in ROS

detoxification[27], suggesting that this ROS increase is a programmed change enacted by *C. glabrata* cells. Indeed, we previously found no evidence of ROS-based oxidative damage to DNA or lipids, suggesting that the ROS produced by *C. glabrata* during echinocandin treatment are not harmful to the cells. Consistent with this conclusion, here we show that deletion of genes involved in ROS detoxification did not reduce *C. glabrata* susceptibility to echinocandins in vitro or inside macrophages and that furthermore, deletion of these genes resulted in a significant increase in the number of ECR colonies. This result suggests that in echinocandin-treated cells ROS in addition to its many effector functions may act as a signal or a stimulant for generating mutations promoting drug resistance. The one intriguing exception to this pattern was the *icl1Δ* mutant that, despite containing increased ROS levels, did not produce any ECR colonies, which may indicate its involvement in multiple aspects of *fks* mutagenesis.

In bacterial literature, the term "tolerance" has emerged to indicate that the entire population is less affected by an antibiotic (e.g., by showing delayed killing dynamics), whereas "persistence" generally refers to a small subset of the population that can survive long-term antibiotic exposure[16]. Both persistence and tolerance lack drug target alterations, and both reproducibly give rise to progenies that exhibit survival/growth phenotypes similar to the parental strains. In general, in medical mycology the terms "tolerance" and "persistence" have been used somewhat interchangeably[27,63,64], and this has been further complicated by the usage of the term "tolerance" for both cidal and static drugs. For the former, tolerance refers to survival in the presence of supra-MIC drug concentrations, whereas for the latter, tolerance refers to incomplete growth inhibition. In this study, we focused on cidal drugs and characterized a subpopulation of cells able to withstand their killing activity and are subsequently able to form colonies on drug-free medium, which we term "persisters" according to bacterial nomenclature. Given that both tolerance and persistence are potentially of clinical importance and that mechanisms underpinning them could be different, appreciation of their differences can inspire detailed molecular studies to better comprehend the biology of drug-refractory fungal infections.

Altogether our results indicated that macrophages constitute a permissive reservoir for development of drug persister *C. glabrata* cells, which facilitate the emergence of stable drug resistance. Finally, amphotericin B was featured as a promising therapeutic potential to significantly reduce the repertoire of micafungin persister and resistant *C. glabrata* cells.

## Methods
The described research complies with all relevant ethical regulations and has been approved by the Hackensack Meridian Health committees for Institutional Biosafety and Institutional Animal Care and Use. We also have permissions to use all mentioned clinical isolates.

### *C. glabrata* strains and growth conditions
We used 8 *C. glabrata* isolates, including 7 clinical isolates and a type-strain CBS138 (ATCC2001), which all had the same MIC values, but belonged to different sequence types and various geographical locations (Supplementary Table 1). The deletant mutants were all derived from the reference strain CBS138. *C. glabrata* cells were grown on yeast-peptone-dextrose (YPD) agar plates overnight at 37 °C and the initial inoculum used for the macrophage infection or evaluation of in vitro stresses were incubated in YPD broth overnight at 37 °C.

### THP1 macrophages and growth conditions
Human acute monocyte leukemia cell line-derived macrophages (THP1; ATCC catalog number TIB-202; Manassas, VA) was used to assess the phagocytosis survival of our *C. glabrata* isolates. RPMI 1640 (Gibco, Fisher Scientific, USA) supplemented with 1% penicillin-streptomycin (Gibco, Fisher Scientific, USA) and 10% heat-inactivated

HFBS (Gibco, Fisher Scientific, USA) was used to grow the THP1 cells. One million THP1 cells treated with 100 nM phorbol 12-myristate 13-acetate (PMA, Sigma) were seeded into 24-well plates and incubated at 37 °C in 5% $CO_2$ for 48 h to induce attachment and differentiation into active macrophages. On the day of infection macrophages were washed with PBS, fresh RPMI was added, and *C. glabrata* cells were exposed to macrophages. Subsequently, the plates were centrifuged ($200 \times g$, 1 min), and incubated at 37 °C in 5% $CO_2$ for 3 h. In the next step, the supernatant RMPI was removed, and the pellets were five times washed with phosphate-buffered saline (PBS) solution. Macrophage lysis was performed by adding 1 ml cold water (kept at 4 °C) and 100 μl of the lysed macrophages were serially diluted, plated on YPD agar, and incubated at 37 °C for up to 72 h.

### Sequencing of HS1 and HS2 of *FKS1* and *FKS2*
HS regions of ECR colonies emerging on YPD plates containing micafungin were amplified and sequenced using primers and conditions described elsewhere. The WT sequences of *FKS1* and *FKS2* from ATCC2001 were used as our control[51].

### Analyzing the effects of antifungal drugs on planktonic or intra-macrophage *C. glabrata*
$10^6$ THP1 macrophages per well were infected with $10^7$ *C. glabrata* cells (MOI of 10/1) and after 3 h washed extensively to remove non-adherent yeast. Fresh RPMI containing the indicated concentrations of antifungal drugs, as well as drug-free RPMI (untreated controls), was added to the infected macrophages and incubated for the indicated amounts of time. At the designated time-points, macrophages were lysed in cold water and the released yeast was plated on YPD-agar to calculate CFU. For planktonic cells, the procedure was the same, except *C. glabrata* was cultured directly in RPMI and no lysis was necessary. For every time-point, the effect of the antifungal drug was calculated as % CFU (treated relative to untreated).

### Measuring the killing dynamic and recovery of ECR colonies following exposure to micafungin
PMA-treated THP1 macrophages were infected with *C. glabrata* isolates belonging to different STs with the MOI of 10/1 (10 *C. glabrata* cells/1 macrophage), which served as our treated groups. Since *C. glabrata* can dramatically replicate inside the macrophages if not treated with micafungin and if incubated for a long time, the THP1 macrophages of the untreated groups were infected with the MOI of 1/10. Three hours post-exposure, macrophages were extensively washed with PBS and the RPMI containing 0.125 μg/ml of micafungin was used for the treated group, while only fresh RPMI was added to the control group. Simultaneously, respective *C. glabrata* isolates were incubated in RPMI containing micafungin and untreated groups were grown in drug-free RPMI. At each time-point 100 μl of the lysate was plated on YPD agar and the rest were centrifuged, the supernatant was decanted, 200 μl of PBS was added, and resuspended cells were transferred to YPD plates containing 0.125 μg/ml of micafungin (from 24-h onward). We chose 0.125 μg/ml of micafungin, since it could detect all the ECR *C. glabrata* isolates harboring various clinically relevant mutations and since the incubation times for the detection of ECR colonies varied depending on mutation type and ECR cell number (we used serial dilution), we extended the incubation time (37 °C) to 7 days to ensure that we could capture all ECR colonies regardless of mutation type and cell number. All treated and untreated groups from both macrophages and RPMI arms were plated on YPD agar containing micafungin to monitor if any ECR colonies could emerge from the untreated groups. The dynamic of killing was measured up to 120 h and CFU of the treated groups were normalized against untreated group at respective time-points. Of note, since the number of *C. glabrata* cells of treated groups were 100 times higher than the untreated counterparts, this dilution factor was considered in our normalization.

## Measuring intracellular concentration of micafungin

THP-1 monocytes grown in supplemented RPMI 1640, were seeded into 96-well tissue culture-treated plates at $5 \times 10^4$ cells/well. THP-1 monocytes were differentiated overnight to macrophages with 100 nM phorbol 12-myristate 13-acetate. Culture medium was replaced with fresh medium containing micafungin, rifampicin, moxifloxacin, or clofazimine. Micafungin was assayed at 3.14 μM and drug controls were assayed at 5 μM. Each drug was tested in triplicate wells. After 0.5, 3, and 24 h incubation at 37 °C, the cells were washed twice with cold PBS to remove extracellular drug. Cells were extracted with 60% DMSO for 1 h at 37 °C. Micafungin concentration in each sample was analyzed by LC-MS/MS, and normalized by the number of cells per well and the average THP-1 cellular volume to calculate intracellular concentrations[65,66]. Intracellular drug accumulation is expressed as the ratio between the intracellular concentration and extracellular concentration (IC/EC).

## LC-MS/MS for intra-macrophage drug concentrations

Neat 500 μM micafungin solution was serially diluted in 50/50 acetonitrile (ACN)/water to create neat standard curves and quality control spiking solutions. 10 μL of neat spiking solutions were added to 90 μL of drug-free macrophage lysate to create standard and QC samples. 10 μL of control, standard, or study sample lysate were added to 100 μL of a 50:50 acetonitrile: methanol protein precipitation solvent mix containing 10 ng/mL of the internal standard verapamil to extract micafungin. Extracts were vortexed for 5 min and centrifuged at $3700 \times g$ for 5 min. 75 μL of supernatant was transferred for LC-MS/MS analysis and diluted with 75 μL of Milli-Q deionized water. LC-MS/MS analysis was performed on a Sciex Applied Biosystems Qtrap 6500+ triple-quadrupole mass spectrometer coupled to a Shimadzu Nexera X2 UHPLC system to quantify each drug concentrations in each sample. Chromatography was performed on an Agilent SB-C8 (2.1 × 30 mm; particle size, 3.5 μm) using a reverse phase gradient. Milli-Q deionized water with 0.1% formic acid was used for the aqueous mobile phase and 0.1% formic acid in acetonitrile for the organic mobile phase. Multiple-reaction monitoring of parent/daughter transitions in electrospray positive-ionization mode was used to quantify all analytes. The MRM transitions of 455.40/165.20, 1270.40/1190.40, 823.5/791.6, 402.2/358.0, 473.2/431.2 were used for verapamil, micafungin, rifampicin, moxifloxacin, and clofazimine, respectively. Data processing was performed using Analyst software (version 1.6.3; Applied Biosystems Sciex).

## Measuring the survival of C. glabrata cells released from macrophages and subsequently exposed to cidal antifungal drugs

THP1 macrophages were infected with MOI of 10/1 and after 3 h they were washed extensively to remove non-adherent cells. Macrophages were lysed using ice-cold water followed by vigorous pipetting and C. glabrata cells were collected using centrifugation. One ml of RPMI 1640 containing micafungin (0.06 μg/ml), caspofungin (0.125 μg/ml), and amphotericin B (2 μg/ml) were added to collected cells and cell suspensions were incubated at 37 °C for 1 h. Prior and 1 h after incubation suspensions were plated on YPD plates, which were incubated at 37 °C for 1 day. The CFU of treated cells were normalized against those prior to exposure and the values were presented as percentage.

## FITC and AF-647 ConA staining

PBS washed initial inoculum of C. glabrata cells were stained with 200 μg/ml of FITC (Millipore Sigma) in carbonate buffer (0.1 M Na$_2$CO$_3$, 0.15 M NaCl, pH = 9.3) and incubated at 37 °C, followed by 3 times PBS washing. The THP1 macrophages were infected with MOI of 10/1 and after 3-, 6-, 24-, and 48-h pst with micafungin (0.125 μg/ml), the C. glabrata cells were released from macrophages and counterstained with 50 μg/ml AF-647-ConA (Millipore Sigma) in PBS buffer+2% BSA. The pellets were washed 3 times with PBS and subjected to flow

cytometry (BD Biosciences). Double-stained yeasts were mother cells, while yeast cells only stained with AF-647 ConA represented replicated cells.

## PI staining

C. glabrata cells released from macrophages were treated PBS containing 10 μg/ml of propidium iodide (PI, Millipore Sigma) and analyzed by FACS (Melody instrument, BD Biosciences).

## Mouse systemic infections

The mouse systemic infection study was approved by the Hackensack Meridian Health Institutional Animal Care and Use Committee protocol number 262 (initial approval date March 18, 2019; expiration date March 17, 2025). Six-week-old CD-1 female mice were used for in vivo systemic infections. Four days prior to infection, mice were immunosuppressed using 150 mg/kg of cyclophosphamide and immunosuppression with 100 mg/kg of cyclophosphamide was continued every 3 days afterward. At the day of infection (day 0), 50 μl of cell suspension containing $5 \times 10^7$ cell was administered intravenously. At 24 h post-infection, 5 mice were sacrificed to measure initial organ burdens. The rest of the mice were grouped into two arms (n = 20), treated daily with 5 mg/kg of caspofungin starting at 24 h after infection and continued until the end of experiment (10 mice), and the second arm served as control, which included mice treated with PBS only (10 mice). Five mice per each group were euthanized and sacrificed at day 3 and 7 (days 2 and 6 post-treatment), spleen and kidneys were harvested and homogenized, and 100 μl of each homogenate was plated on YPD plate and incubated for 24–48 h at 37 °C. The number of colonies from each organ of treated mice were normalized against the respective organ of untreated group and data presented as percentage in Fig. 2b.

## LC-MS/MS of caspofungin in mouse tissue

1 mg/mL stock of Caspofungin was serially diluted in 50/50 acetonitrile (ACN)/water to create neat standard curves and quality control spiking solutions. Mouse kidney and spleen tissues and drug-free control mouse tissues were weighed and homogenized in 4 volumes of PBS to a final 5x dilution factor. Homogenization was achieved using a FastPrep-24 instrument (MP Biomedicals) and 1.4 mm zirconium oxide beads (Bertin Corp.). 10 μL of neat spiking solutions were added to 90 μL of drug-free kidney and spleen homogenates to create standard and QC samples. 10 μL of control, standard, or study sample lysate were added to 100 μL of a 50:50 acetonitrile: methanol protein precipitation solvent mix containing 10 ng/mL of the internal standard verapamil to extract caspofungin. Extracts were vortexed for 5 min and centrifuged at 4000 RPM for 5 min. 75 μL of supernatant was transferred for LC-MS/MS analysis and diluted with 75 μL of Milli-Q deionized water. LC-MS/MS analysis was performed on a Sciex Applied Biosystems Qtrap 6500+ triple-quadrupole mass spectrometer coupled to a Shimadzu Nexera X2 UHPLC system to quantify each drug concentrations in each sample. Chromatography was performed on a Luna Omega C18 column (2.1 × 100 mm; particle size, 3 μm) using a reverse phase gradient. Milli-Q deionized water with 0.1% formic acid was used for the aqueous mobile phase and 0.1% formic acid in acetonitrile for the organic mobile phase. Multiple-reaction monitoring of parent/daughter transitions in electrospray positive-ionization mode was used to quantify all analytes. The MRM transitions of 455.40/165.20 and 547.41/538.4 were used for verapamil and caspofungin, respectively. The double-charged ion was used for caspofungin. Data processing was performed using Analyst software (version 1.6.3; Applied Biosystems Sciex).

## Generation of C. glabrata knock-out mutants

The following genes were subjected to deletion, catalase 1 (CAT1, CAGL0K10868g), glutathione oxidoreductase (GRX2, CAGL0K05813g), manganese superoxide dismutase (SOD2, CAGL0E04356g), and three

transcription factors playing role in oxidative stress responses, namely *SKN7* (CAGL0F09097g), *MSN4* (CAGL0M13189g), and *YAP1* (CAGL0H04631g), and *ICL1* (CAGL0L09273g). Knock-out mutants were created in house using a previously described protocol[65], in which the open reading frame of the gene of interest was replaced by nourseothricin (NAT) resistance cassette. The knock-out construct was generated by using Ultramer primers (~80–100 bps) containing homology regions with NAT and with regions flanking GOIs. Competent cells were created by log-phase grown *C. glabrata* cells using Frozen-EZ Yeast Transformation Kit (Zymo Research) and transformation followed an electroporation-based protocol described previously[67]. The colonies growing on YPD plates containing NAT were subjected to PCR and sequencing using diagnostic primers listed in Supplementary Table 3 (the primers used in the current study were manufactured by Integrated DNA Technologies and Sanger Sequencing carried out by Genewiz). Two independent knock-out mutants were used for each experiment.

### Measuring metabolic dependency of antifungal drugs

Overnight grown *C. glabrata* cells (CBS138) were PBS washed thrice, enumerated, and 100 μl of $10^8$ cells were added to 1 ml of various RPMI concentrations containing micafungin or caspofungin or amphotericin B. Desired concentrations of RPMI were simply made by addition of sterilized demi water to 100% RPMI. The survival of *C. glabrata* cells incubated in various 100%-, 20%-, and 2%-RPMI containing micafungin (0.125 μg/ml) or caspofungin (0.25 μg/ml) or amphotericin B (2 μg/ml) was assessed at 3-, 6-, and 24-h pst (totaling 9 conditions for each time-point). *C. glabrata* cells incubated in respective RPMI concentrations lacking any antifungal drugs were considered as control. The survival rate of drug exposed *C. glabrata* cells were normalized against drug-free controls, which was presented by percentage. A higher metabolic dependency was defined when the lethality of a given antifungal drug significantly decreased at lower concentrations of RPMI.

### Measuring the impact of amphotericin B or micafungin treatment on ICG survival rate

ICG cells (CBS138) treated with micafungin (0.125 μg/ml) for 24 h were exposed to 1 ml of fresh RPMI containing either amphotericin B (2 μg/ml) or micafungin (0.125 μg/ml) and the survival rate was assessed 24 and 48 h later. The survival rates of treated ICG were normalized against the counterparts treated with drug-free RPMI and the data were presented by percentage.

### Measuring the impact of amphotericin B or micafungin treatment on ECR rate

Overnight grown knockout mutants, *cta1Δ*, *yap1Δ*, and *skn7Δ*, were PBS washed twice, enumerated, and 100 μl of $10^8$ cells were added to 1 ml of 100% RPMI containing micafungin (0.125 μg/ml) and incubated for 24 h. After 24 h the *C. glabrata* cells were collected, washed with PBS, and either treated with fresh RPMI containing micafungin (0.125 μg/ml) or amphotericin B (2 μg/ml) and survival rate and mutation frequency were assessed 24 and 48 h later. The respective *C. glabrata* mutants grown in drug-free RPMI were used as control.

### ROS measurement

ROS was measured using dihydroethidium (DHE) (6.5 μg/ml; ThermoFisher), which detects superoxide levels[27,68,69]. *C. glabrata* isolates lacking *CTA1*, *GRX2*, *ICL1*, *MSN4*, *SKN7*, *SOD2*, and *YAP1* and the parental WT strain of CBS138 were incubated in RPMI (Gibco™ RPMI 1640 Medium) containing 0.125 μg/ml of micafungin for 2 h. CBS138 incubated in drug-free served as our negative control. Since our previous study using the same dyes showed the lack of ROS detection from ECR isolates, we did not include ECR isolates in this experiment[27]. After each time-point, *C. glabrata* cells, at least in three biological replicates,

were collected, washed with prewarmed PBS, and stained with DHE in PBS for 30–40 min at 37 °C. Subsequently, the stained cells were collected, 1 ml of prewarmed PBS was added and subjected to flow cytometry using BD FACS Diva and the data were analyzed by FlowJo software v10.6.1 (BD Biosciences).

### Antifungal susceptibility testing (AFST)

The broth microdilution protocol of the CLSI M27-A3 was followed. AFST included the following antifungal drugs, fluconazole (Pfizer), amphotericin B (Sigma-Aldrich), micafungin (Astellas Pharma), and anidulafungin (Pfizer). Plates were incubated at 37 °C for 24 h, and the MIC50 data (50% growth reduction compared to controls without drug) were determined visually. Each experiment included at least three biological replicates.

### Statistical analysis methods

Statistical analysis was performed by SPSS v.24 (SPSS Inc) and GraphPad Prism v.9 and values ≤0.05 were considered as statistically significant. Data presented in Fig. 2 used paired t-test, while those shown in Figs. 3 and 6 were analyzed by modified t-test. Modified and paired t-tests were used to analyze the data shown in Fig. 5. Statistical analyses were performed using JMP®, Version 16.2.0 (SAS Institute Inc., Cary, NC). Where sample size allowed ($n > 3$), normality was checked visually through a normal quantile plot and goodness of fit test using Shapiro–Wilk. Otherwise, data were assumed to be normally distributed. A significance level of 0.05 was used to determine if the results were statistically significant. Parametric analysis was performed using paired T-test, pooled T-test, or modified T-test, where appropriate, while nonparametric analysis was performed using Wilcoxon signed-rank test.

### Reporting summary

Further information on research design is available in the Nature Portfolio Reporting Summary linked to this article.

## Data availability

This paper contains no data with mandated deposition. All strains generated as part of this study will be available upon request. Source data are provided with this paper.

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

## Acknowledgements

We would like to thank Enriko Dolgov and Steven Park for assistance with the mouse sepsis model, Firat Kaya for assistance with measuring caspofungin concentrations in mouse tissues, and Tara Lozy for help with statistical analysis. This work was supported by NIH 5R01AI109025 to D.S.P. The Flow Cytometry & Cell Sorting Shared Resource was supported by NIH/NCI grant P30-CA051008.

## Author contributions

A.A. and F.D. designed the study, performed most of the experiments, and analyzed the data. E.S. and D.S.P. analyzed the data and provided supervision and guidance. N.C. performed the animal model experiments. J.S., M.Z., and S.P.L. performed the LC-MS/MS analyses. A.A., F.D., and E.S. wrote the manuscript. D.S.P. and J.S. edited the manuscript.

## Competing interests

D.S.P. serves on the advisory board of Scynexis.
