## [Peer review file · Nature Communications]

REVIEWER COMMENTS

Reviewer #1 (Remarks to the Author):

This study illustrates the response of *C. glabrata* when exposed to both the challenges of drug exposure and the intracellular environment of macrophages. The authors show that persister cells can be produced in these conditions. The authors also suggest that the production of ROS in macrophages stimulates the formation of persister cells and that genes involved in resistance to ROS is critical in this step. Importantly, the formation of persister cells is linked with emergence of candidin-resistant cells. This emergence and the formation of persister cells can be prevented by alternate treatment with amphotericin B, thus suggesting that this drug could be used to diminish persistent reservoirs. In conclusion, this study provides novel insights into the existence of persister cells in *C. glabrata*. Some questions remain:

- 1) The authors refer to persister cells as those able to survive in a cell population in the presence of a drug, they stated in the Discussion: "In this study we focused on cidal drugs and characterized a sub-population of cells able to withstand their killing activity, which we term "persisters" according to bacterial nomenclature". Persistence is therefore based on the analysis of a cell population, and this has not been carried out here. Here authors only looked at the whole set of cells in their survival assays. They state in their discussion "tolerance refers to survival in the presence of supra-MIC drug concentrations". I think some rephrasing is needed early in the study. One can state that tolerance to drugs leads to the selection of persister cells for example.
- 2) Fig 1c: authors mention "biphasic" killing upon macrophages exposure. How is the killing of planktonic cells looking? No biphasic killing in these cells?
- 3) Authors used both caspofungin and micafungin in their experiments. For example, in Fig 2b, caspofungin is used while in Fig 2d, micafungin is used to show the degree of survival after macrophages internalization. In Fig 2e, caspofungin is used in animal experiments. There are several instances in which authors used either caspofungin and micafungin. What are the reasons for this drug regimen changes? It would be advisable to keep the same type of drug throughout the experiments.
- 4) The conclusions linked to Fig 2e "...indicating that macrophage-rich anatomical niches likely contain drug-tolerant persister *C. glabrata* cells" can be still be challenged by other factors, for example by drug distribution in specific organs (such as the spleen). I think the conclusion is somehow premature unless it can be shown that the drug concentration is not organ-dependent.
- 5) ROS production by macrophages: authors tend to suggest that ROS production by macrophages is linked to drug tolerance via decrease of ATP production and reduction of translation/transcription in *C. glabrata*. The current data cannot really convince about the causes of drug tolerance in *C. glabrata* via specific macrophage-dependent factors. The only evidence provided here is that a chemically-induced effect (use of an NADPH oxidase inhibitor) decreases to some extent drug tolerance in *C. glabrata*. Decrease of ATP production can be the result of decreased transcription and translation (which reflects

growth inhibition and therefore decreased cell metabolism reflected by low ATP levels) that is itself not necessarily dependent on the production of ROS by macrophages. When the authors state that “We hypothesized that the observed macrophage-induced down-regulation of transcription and translation could be due to a reduction in ICG ATP stores...”, there are no data suggesting that ATP levels control the extent of transcription and translation. The cause/effect relationships between ROS production, ATP levels, transcription and translation are difficult to establish here. This is a weak part of the study that needs to be removed or modified from the current manuscript.

6) When mentioning “ECR frequencies”, authors refer to the number of detected resistant isolates after plating on drug-containing medium. Can authors guess the mutation rates in ICG versus planktonic cells?

7) Are also ICG cells giving rise to resistance to other types of drugs such as azoles and/or amphotericin B? One may expect that mutation frequencies caused by the macrophage environment can select for drug resistance types different from cansin resistance.

8) Fig 2 labels are not correctly mentioned in the text.

Reviewer #2 (Remarks to the Author):

This study is aimed at supporting the hypothesis that *Candida glabrata* acts as a reservoir of drug resistant infection inside macrophages and at proposing new strategies to eliminate those persisters. Globally the manuscript is well written and the topic is of great interest. Moreover, the authors have already proved their expertises for the in vitro characterization of *C. glabrata* persisters so the study is really well conducted. However, some points need to be discussed and/or specified before the manuscript be suitable for publication.

1 - Concerning the methods used for comparison between untreated/treated yeast. The authors decided to normalize their results using the untreated condition using a yeast concentration (in terms of MOI) 100 less than in the treated samples. Thus, how can the authors be sure that this huge difference does not impact metabolic pathways induced in macrophages? How can they be sure that the two phenomenons observed are comparable?

2 - Concerning the drug concentrations used all along the study, can the authors explain their different choices? Why, in some experiments, they use 0.125µg/ml, while, in others they used 0.06µg/ml? What are the MIC values of each antifungal drug used in the study for each yeast strain? (I couldn't find the supplemental data which mention MIC values). Same way, why some experiments use 2X MIC when others use 4X MIC? Again, how can we be sure that variations in the drug concentrations used do not impact the macrophage's responses and/or yeast modifications? Do the concentrations used are

comparable to the expected patient doses? The authors have to discuss their in vitro results in comparison to human medical practices. Does the phenomenon observed in vitro could be translated to humans? Do people who received prophylactic antifungal drugs may be more able to present ICG persisters and so present higher risk for invasive candidiasis?

3 - Concerning the mice model, it would be important to show mice survival curves as well as to perform anatomopathological analyses of the spleen and the kidney in order to study the pathophysiological impact of ICG. Moreover, the authors used BYP40 strain, which is not the same strain than those used in the in vitro experiments. Please justify why? How can we be sure that the different strains act in the same way? Indeed, a growing body of literature shows that *Candida* phenotypic heterogeneity correlates with different host responses.

4 - Concerning chapter related to fig 2: Do the authors tried to pre-exposed strains to antifungal treatment before using them for macrophages infection? Do the pretreated yeast act differently compared to naïve cells? If they do not, it will be interesting to perform it in order to study the impact of drug exposure upon yeast metabolism and so upon immune response. Indeed, antifungal drug exposure of *Candida* cells modifies the cell wall composition and architecture and so can modulate the host response. In addition, in humans, the use of prophylactic antifungal drugs exposes the colonizing yeasts of the human microbiota before the occurrence of invasive fungal disease and so may influence the occurrence of ICG.

- Fig 1c: why did the authors select AmphoB for testing ICG persisters obtained from a micafungin treatment? Do the authors' hypothesis is that there is a non-drug specific persistence effect in macrophages? Please specify.

- Fig1: the authors talk about a pilot study they performed. Does this pilot study have been published? If yes please add the reference. If not please add methods and results in supplemental data.

- Fig 1g and paragraph related-to: should be supplemental data if the manuscript is too long.

- Fig 2 is not well cited in the main manuscript. (Fig 2a>2b; 2b>2c ...). Please correct

- Fig 4: please specify the % of FKS2 mutation found in ECR colonies from planktonic forms compared to ECG from ICG.

- The statistical analysis chapter is lacking in the mat and meth section. Please add.

Reviewer #3 (Remarks to the Author):

Comments

In this paper the authors have tested the efficacy of echinocandins against intracellular *Candida glabrata* and explored the emergence of echinocandin-resistance within the population of intracellular yeasts.

The study is interesting because of the frequency of *C. glabrata* infections and the difficulties in treating these infections, sometimes related to the emergence of echinocandin resistance in this species.

Overall, the results showed that after exposure to micafungin, *C. glabrata* has better survival inside macrophages compared to planktonic conditions and that the intracellular reservoir promotes the emergence of micafungin-resistant fks mutants. Moreover, the study showed that ROS produced by macrophages seems to be the key element to the emergence of echinocandin-resistant mutants.

Major points

1. There is a certain degree of killing of planktonic cells by fluconazole (about 90%) and voriconazole (about 98%). This is unexpected for these fungistatic drugs. How can these results be explained? On Figure 1a, killing by fluconazole seems equivalent to killing by micafungin which is surprising for a fungistatic agent. Moreover, the curves for planktonic conditions for these two drugs seem almost identical.
2. In the in vivo experiment, mice were infected with BYP40. Why did the authors not use one of the strains previously used for in vitro experiments?
3. After 48h, micafungin-treated planktonic cells displayed a CFU rebound (Figure 4) but no resistant mutants were detected in this growth condition. How this growth rebound could be explained if there are no resistant mutants?
4. In the experiments exploring the frequency of ECR emergence in mutants (Figures 5e and Supplementary Table 2), it seems that less mutants were obtained for the WT strain compared to the experiment presented on Figure 4. Are these differences significant?
5. For the mutation type and frequency of ECR colonies derived from the planktonic condition, there are no data for the WT strain (Supplementary table 2). This makes comparisons with the results for mutants difficult.
6. In the Introduction and Discussion it is stated that "Yet, it has not been determined how intracellular *C. glabrata* responds to antifungal drugs". Nevertheless, there are some previous studies that explore the activity of antifungals against *C. glabrata* inside macrophages (for example: Baltch AL, et al. Time-kill studies with micafungin and voriconazole against *Candida glabrata* intracellularly in human monocyte-derived macrophages and extracellularly in broth. *Diagn Microbiol Infect Dis.* 2011 Aug;70(4):468-74.; Bopp LH, et al. Antifungal effect of voriconazole on intracellular *Candida glabrata*, *Candida krusei* and *Candida parapsilosis* in human monocyte-derived macrophages. *J Med Microbiol.* 2006 Jul;55(Pt 7):865-870). Results of these previous works should be discussed.

Minor points

1. Introduction: The authors list a group of intracellular bacteria. It should be noted that *Staphylococcus aureus* is generally not considered as a facultative intracellular bacterium.
2. Results: The MIC of the four drugs could be indicated in the text.
3. Figure 1b: In this experiment a 6 log decrease can be detected (100 to 0.0001). Please indicate in the Material and Methods, the number of yeast cells inoculated in each well and the limit of detection for CFU determination.
4. Figure 2: The different panels cited in the text do not match with the Figure, both for in vitro and in vivo experiments.

Reviewer #1

1) The authors refer to persister cells as those able to survive in a cell population in the presence of a drug, they stated in the Discussion: "In this study we focused on cidal drugs and characterized a sub-population of cells able to withstand their killing activity, which we term "persisters" according to bacterial nomenclature". Persistence is therefore based on the analysis of a cell population, and this has not been carried out here. Here authors only looked at the whole set of cells in their survival assays. They state in their discussion "tolerance refers to survival in the presence of supra-MIC drug concentrations". I think some rephrasing is needed early in the study. One can state that tolerance to drugs leads to the selection of persister cells for example.

Indeed, we (and the bacterial literature) define "persisters" as a subset of the population (inside the host or in culture) that are able to survive exposure to cidal drugs. Thus, survival, measured as the ability to form colonies (while the majority of the population is killed), is a defining feature of this sub-population, and this is what we analyzed in this study. We also defined other features of *C. glabrata* persisters, such as their non-proliferating state and susceptibility to amphotericin B. The relationship between "tolerance" and "persistence" is on the one hand straightforward, in that increased tolerance, such as under conditions of stress, leads to increased numbers of persisters, and vice versa, but also confusing in that the two terms are frequently used interchangeably. Indeed, persisters are by definition more tolerant of cidal drugs than non-persisters to be able to survive at supra-MIC concentrations. We clarify these points in the Discussion (lines 356-366 in the marked-up manuscript).

2) Fig 1c: authors mention "biphasic" killing upon macrophages exposure. How is the killing of planktonic cells looking? No biphasic killing in these cells?

In planktonic cells the killing is also biphasic but the surviving population is much smaller than in intra-macrophage cells. This can be seen in Figure 2d.

3) Authors used both caspofungin and micafungin in their experiments. For example, in Fig 2b, caspofungin is used while in Fig 2d, micafungin is used to show the degree of survival after macrophages internalization. In Fig 2e, caspofungin is used in animal experiments. There are several instances in which authors used either caspofungin and micafungin. What are the reasons for this drug regimen changes? It would be advisable to keep the same type of drug throughout the experiments.

Caspofungin and micafungin are the most highly prescribed echinocandin drugs and are used widely in the clinic. We used both caspofungin and micafungin in several *C. glabrata*-macrophage experiments, and the results were always very similar. An example of this can be seen by comparing Figure 2b, 3h post-infection (micafungin) with Figure 2c (caspofungin, also 3h post-infection). Furthermore, our recent paper (Garcia-Rubio et al., mBio 2021) showed that caspofungin and micafungin treatments in vitro elicited very similar transcriptional programs in *C. glabrata* persister cells. Since performing all experiments with both drugs would be extremely laborious, time-consuming, and costly, we focused our in-depth investigation on micafungin. However, we used caspofungin for the mouse experiment because the mouse caspofungin treatment protocol is well-established in the Perlin lab. Finally, both Figure 2b and 2d show results obtained with micafungin. This was correctly indicated in the figure and in the legend but was made confusing in the text due to the mix-up with panel labels (2a instead of 2b, etc.). It has now been corrected.

4) The conclusions linked to Fig 2e "...indicating that macrophage-rich anatomical niches likely contain drug-tolerant persister *C. glabrata* cells" can still be challenged by other factors, for example by drug distribution in specific organs (such as the spleen). I think the conclusion is somehow premature unless it can be shown that the drug concentration is not organ-dependent.

We thank the reviewer for this suggestion. As we have repeated the animal model using strain CBS138, we also obtained organ samples from treated mice on days 2 and 6 after the start of treatment and measured caspofungin concentrations. As we show in the revised manuscript (new Supplementary Figure 1b), caspofungin effectively accumulates in both organs, and, although caspofungin concentration in the spleen is about 2-fold lower than that in the kidney, both concentrations greatly exceed the MIC. For instance, by day 6 of treatment the spleen caspofungin concentration exceeds the MIC by 48-fold, yet there is no evidence of the drug's efficacy in this organ. As we state in the revised manuscript (lines 184-187 of the marked-up manuscript), our data support the conclusion that caspofungin is less active against *C. glabrata* in the spleen than in the kidney, which is likely due to the high abundance of intra-macrophage *C. glabrata* in the former, but may also be influenced by somewhat lower drug penetration.

5) ROS production by macrophages: authors tend to suggest that ROS production by macrophages is linked to drug tolerance via decrease of ATP production and reduction of translation/transcription in *C. glabrata*. The current data cannot really convince about the causes of drug tolerance in *C. glabrata* via specific macrophage-dependent factors. The only evidence provided here is that a chemically-induced effect (use of an NADPH

oxidase inhibitor) decreases to some extent drug tolerance in *C. glabrata*. Decrease of ATP production can be the result of decreased transcription and translation (which reflects growth inhibition and therefore decreased cell metabolism reflected by low ATP levels) that is itself not necessarily dependent on the production of ROS by macrophages. When the authors state that “We hypothesized that the observed macrophage-induced down-regulation of transcription and translation could be due to a reduction in ICG ATP stores...”, there are no data suggesting that ATP levels control the extent of transcription and translation. The cause/effect relationships between ROS production, ATP levels, transcription and translation are difficult to establish here. This is a weak part of the study that needs to be removed or modified from the current manuscript.

We agree with this comment and have therefore removed panel 3e and all corresponding passages.

6) When mentioning “ECR frequencies”, authors refer to the number of detected resistant isolates after plating on drug-containing medium. Can authors guess the mutation rates in ICG versus planktonic cells?

In order to calculate mutation rates, a number of conditions have to be met by the experimental system, which are not met here. Most notably, cell death should be negligible under the experimental conditions, which is not the case for *C. glabrata* engulfed by macrophages and exposed to echinocandins. Furthermore, the growth rates of mutants and non-mutants should be the same, which is also not the case for *fks* mutants in the presence of echinocandins. Thus, unfortunately, mutation rates cannot be accurately measured and therefore, we only discuss these results in terms of mutation frequencies.

7) Are also ICG cells giving rise to resistance to other types of drugs such as azoles and/or amphotericin B? One may expect that mutation frequencies caused by the macrophage environment can select for drug resistance types different from candidin resistance.

This is an interesting and important question, and the prediction would be that, indeed, other mutations and types of genetic instability would also arise in the macrophage environment. We are pursuing this line of investigation in a separate study.

8) Fig 2 labels are not correctly mentioned in the text.

Corrected.

Reviewer #2

1 - Concerning the methods used for comparison between untreated/treated yeast. The authors decided to normalize their results using the untreated condition using a yeast concentration (in terms of MOI) 100 less than in the treated samples. Thus, how can the authors be sure that this huge difference does not impact metabolic pathways induced in macrophages? How can they be sure that the two phenomena observed are comparable?

We believe this question refers to the experiment shown in Figure 4, where *C. glabrata* was used to infect macrophages, exposed to micafungin (or left untreated in the control), and then both total CFU and number of ECR mutants were monitored over a 5-day period. This was the only experiment where the MOI was different between treated and untreated conditions. The reason we used the low MOI for the untreated condition in this experiment was that in the absence of drug, *C. glabrata* rapidly proliferates inside the macrophages and is expected to “overwhelm” and kill the macrophages during the 5-day time course. At the same time, we could not use the low MOI for the treated condition because that would not give us the necessary dynamic range to detect the rare ECR mutants emerging under these conditions. We believe that this was a valid compromise because: (a) under the untreated condition the *C. glabrata* cells multiply inside macrophages and also reach high numbers, making it comparable to the treated condition, and (b) the focus of this experiment was not on studying the macrophage state but the emergence of ECR mutations in the ICG. Moreover, the intracellular replication rate of *C. glabrata* cells remains very similar when using different MOIs. For instance, the replication rate of MOI 1/10 and MOI 1/1 is very similar at 24 hrs (4-6-fold increase for both).

2 - Concerning the drug concentrations used all along the study, can the authors explain their different choices? Why, in some experiments, they use 0.125µg/ml, while, in others they used 0.06µg/ml? What are the MIC values of each antifungal drug used in the study for each yeast strain? (I couldn't find the supplemental data which mention MIC values). Same way, why some experiments use 2X MIC when others use 4X MIC? Again, how can we be sure that variations in the drug concentrations used do not impact the macrophage's responses and/or yeast modifications? Do the concentrations used are comparable to the expected patient doses? The authors have to discuss their in vitro results in comparison to human medical practices. Does the phenomenon observed in vitro could be translated to humans? Do people who received prophylactic antifungal drugs may be more able to present ICG persists and so present higher risk for invasive candidiasis?

In the first experiment (Fig. 1a) we used the lowest drug concentrations (2X MIC), but then, focusing on

micafungin, we showed that the observed phenomenon, i.e., the increased drug tolerance of ICG relative to planktonic cells, was observed over a wider range of concentrations (2X-256X, Fig. 1b). In most subsequent experiments we used the 4X or 8X MIC concentration of micafungin, guided by our observations that ICG was significantly more tolerant than *C. glabrata* cultured in RPMI. When we measured the killing of *C. glabrata* cells released from macrophages, we used 4X MIC, because these cells were no longer intracellular and were more susceptible to killing. The concentrations used in this study do not adversely affect macrophage function. Indeed, our data show that the macrophages are active and kill *C. glabrata* synergistically with the drugs (please see Figures 1b and 4b).

3 - Concerning the mice model, it would be important to show mice survival curves as well as to perform anatomopathological analyses of the spleen and the kidney in order to study the pathophysiological impact of ICG. Moreover, the authors used BYP40 strain, which is not the same strain than those used in the in vitro experiments. Please justify why? How can we be sure that the different strains act in the same way? Indeed, a growing body of literature shows that *Candida* phenotypic heterogeneity correlates with different host responses.

The *C. glabrata* sepsis model does not result in animal death during the examined time period, and therefore survival is not an endpoint in these experiments. We have repeated the model using the same strain as was used in the *in vitro* and macrophage experiments, CBS138, and obtained congruent results, which are shown in the revised manuscript (Fig. 2b).

4 - Concerning chapter related to fig 2: Do the authors tried to pre-exposed strains to antifungal treatment before using them for macrophages infection? Do the pretreated yeast act differently compared to naïve cells? If they do not, it will be interesting to perform it in order to study the impact of drug exposure upon yeast metabolism and so upon immune response. Indeed, antifungal drug exposure of *Candida* cells modifies the cell wall composition and architecture and so can modulate the host response. In addition, in humans, the use of prophylactic antifungal drugs exposes the colonizing yeasts of the human microbiota before the occurrence of invasive fungal disease and so may influence the occurrence of ICG.

The effects of echinocandins on fungus-macrophage interactions have been extensively studied, and, in particular, the experiments suggested by the reviewer have been published by the Munro lab (Walker and Munro 2020; PMID: 32528900). The authors reported that in *C. glabrata* pretreatment with caspofungin had minor or no effects on glucan, mannan, and chitin levels and did not alter yeast uptake by macrophages or macrophage production of TNF α . Overall, the study showed that the relationship between yeast cell surface carbohydrate exposure and macrophage uptake is not straightforward, as some strains/species were taken up less by macrophages after pre-exposure to caspofungin, despite having increased beta-glucan and chitin levels. How prophylactic use of antifungal drugs influences the colonizing yeasts of the human microbiota is a very important and interesting question and will undoubtedly be the topic of many future studies.

- Fig 1c: why did the authors select AmphoB for testing ICG persisters obtained from a micafungin treatment? Do the authors' hypothesis is that there is a non-drug specific persistence effect in macrophages? Please specify.

Both our experiments and bacterial literature indicate that a non-proliferating phenotype is a key feature of persisters, which renders them phenotypically resistant to certain types of antifungal agents, particularly drugs whose killing activity depends on the metabolic state of the cell and its proliferative status. Our experiments showed that whereas echinocandin killing activity was strongly influenced by the fungal metabolic state/growth and proliferation, this is not the case for polyene drugs like amphotericin B, which bind to ergosterol and forms pores independent of metabolic state. Thus, we hypothesized that ICG persisters formed during micafungin treatment may be more susceptible to amphotericin B, which was indeed the case. We explain this in the text, lines 283-286 of the marked-up manuscript.

- Fig1: the authors talk about a pilot study they performed. Does this pilot study have been published? If yes please add the reference. If not please add methods and results in supplemental data.

We have added the results of the pilot study in the new Supplementary Table 2. We used clinically derived ECR *C. glabrata* isolates harboring various mutations in HS1-Fks1 (S629P and R631G) and HS1-Fks2 (F659V, F659Y, S663P, and S663F) and plated 10³-10⁶ CFUs on YPD plates containing 0.125 μ g/ml or 0.5 μ g/ml of micafungin. Plates were incubated at 37C for up to 7 days and the number of colonies emerging on micafungin containing YPD plates were recorded every day and compared it to the expected number, which was based on the number of plated cells. Generally, for the YPD plates containing 0.125 μ g/ml of micafungin, there was a good concordance between expected and obtained CFU, and the time to colony emergence and the colony number varied depending on the mutation, with those carrying R631G in Fks1 producing a lower colony number, especially at lower cell numbers (\leq 1000). On the other hand, the number of colonies emerging on YPD plates containing 0.5 μ g/ml of micafungin decreased dramatically. Therefore, to detect the ECR colonies

with a higher sensitivity and specificity, we used YPD plates containing 0.125µg/ml of micafungin. Of note, echinocandin susceptible counterparts did not yield colonies on plates containing either concentration.

- Fig 1g and paragraph related-to: should be supplemental data if the manuscript is too long. We will move the figure and corresponding text to supplementary data, if so advised by the journal's editorial team.

- Fig 2 is not well cited in the main manuscript. (Fig 2a>2b; 2b>2c ...). Please correct
Corrected.

- Fig 4: please specify the % of FKS2 mutation found in ECR colonies from planktonic forms compared to ECR from ICG.

All ECR colonies produced both by planktonic cells and by ICG contained *fkf2* mutations, as we specify in line 240 of the marked-up manuscript. The identity of each mutation is shown in the new Supplementary Table 3. If the question is asking how much less *FKS2* mutagenesis there was in planktonic cells relative to ICG, that is a difficult question to answer precisely because, as we reply to Reviewer 1 above, it is not possible to calculate mutation rates under these conditions.

- The statistical analysis chapter is lacking in the mat and meth section. Please add.

As we state in the revised methods section, statistical analysis was performed by SPSS v.24 (SPSS Inc) and values ≤ 0.05 were considered as statistically significant. Data presented in Figure 2 used paired-t test, while those shown in Figures 3 and 6 were analyzed by modified t-test. Modified and paired t-tests were used to analyze the data shown in Figure 5.

Reviewer #3

1. There is a certain degree of killing of planktonic cells by fluconazole (about 90%) and voriconazole (about 98%). This is unexpected for these fungistatic drugs. How can these results be explained? On Figure 1a, killing by fluconazole seems equivalent to killing by micafungin which is surprising for a fungistatic agent. Moreover, the curves for planktonic conditions for these two drugs seem almost identical.

We apologize for the confusion, which was caused by imprecise labeling of the Y-axes in Figure 1a and which we have corrected. We measured the CFU in treated ICG or planktonic cells and normalized that number to the CFU of untreated samples at the same time-points. Thus, the drop in this number (CFU % treated relative to untreated) can be due either to a lack of growth during drug exposure (for azoles) or to increased killing during drug exposure (for echinocandins and amphotericin B). We make this clearer in the Methods (lines 399-406) and the text (lines 90-92).

2. In the in vivo experiment, mice were infected with BYP40. Why did the authors not use one of the strains previously used for in vitro experiments?

We have repeated the model using the same strain as was used in the *in vitro* and macrophage experiments, CBS138, and obtained very similar results, which are shown in the revised manuscript (Fig. 2b).

3. After 48h, micafungin-treated planktonic cells displayed a CFU rebound (Figure 4) but no resistant mutants were detected in this growth condition. How this growth rebound could be explained if there are no resistant mutants?

This rebound is not due to acquired genotypic resistance but to a phenomenon similar to phenotypic resistance, which has been described for bacteria exposed to cidal antibiotics, wherein a subset of the population starts proliferating in the presence of the drug, although no genetic determinants of resistance (i.e., resistant mutants) are present; and this phenotype is not heritable (PMID: 27029301). The basis for phenotypic resistance is not clear, but it has been hypothesized to be caused by factors such as "a reduction in the antibiotic concentration at high cell densities due to its binding to cell envelopes and debris of alive and killed cells" (PMID: 27029301). Likewise, echinocandins may bind to fragments of membranes and cell wall left from the extensive and rapid early cell killing, decreasing the effective concentration of the drug and allowing the surviving cells to resume proliferating. We address this and add possible explanations in lines 237-240 of the marked-up manuscript.

4. In the experiments exploring the frequency of ECR emergence in mutants (Figures 5e and Supplementary Table 2), it seems that less mutants were obtained for the WT strain compared to the experiment presented on Figure 4. Are these differences significant?

The difference between the number of ECR colonies in Figure 4b and the WT in Figure 5e is that 4b shows the cumulative number obtained from six different strains of *C. glabrata*, including CBS138, whereas 5e shows

results from CBS138 only, which is the parent strain for the deletion mutants. The numbers for CBS138 alone are consistent between the two experiments, as can be seen in new supplementary tables 3 and 4.

5. For the mutation type and frequency of ECR colonies derived from the planktonic condition, there are no data for the WT strain (Supplementary table 2). This makes comparisons with the results for mutants difficult. As we clarify in the Supplementary Table 4 (formerly Supplementary Table 2), the WT strain did not yield any ECR colonies in planktonic cultures.

6. In the Introduction and Discussion it is stated that "Yet, it has not been determined how intracellular *C. glabrata* responds to antifungal drugs". Nevertheless, there are some previous studies that explore the activity of antifungals against *C. glabrata* inside macrophages (for example: Baltch AL, et al. Time-kill studies with micafungin and voriconazole against *Candida glabrata* intracellularly in human monocyte-derived macrophages and extracellularly in broth. *Diagn Microbiol Infect Dis.* 2011 Aug;70(4):468-74.; Bopp LH, et al. Antifungal effect of voriconazole on intracellular *Candida glabrata*, *Candida krusei* and *Candida parapsilosis* in human monocyte-derived macrophages. *J Med Microbiol.* 2006 Jul;55(Pt 7):865-870). Results of these previous works should be discussed.

We thank the reviewer for pointing out these studies. We now rephrase the previous inaccurate sentence in the Introduction, citing Balch et al. (line 70 in the marked-up manuscript), and delete the similar sentence in Discussion. We are not citing Bopp et al. because they only focused on voriconazole and only examined its effect on ICG without comparing it to planktonic cells.

Minor points

1. Introduction: The authors list a group of intracellular bacteria. It should be noted that *Staphylococcus aureus* is generally not considered as a facultative intracellular bacterium.

We have rephrased this sentence and no longer refer to *S. aureus* as an intracellular bacterium (lines 52-54 in the marked-up manuscript).

2. Results: The MIC of the four drugs could be indicated in the text.

The MIC information for all the strains has been added as new Supplementary Table 1 and referred to in the text (line 83 in the marked-up manuscript).

3. Figure 1b: In this experiment a 6 log decrease can be detected (100 to 0.0001). Please indicate in the Material and Methods, the number of yeast cells inoculated in each well and the limit of detection for CFU determination.

As we now indicate in the Methods sections (line 401 of the marked-up manuscript), 10^7 yeast cells were added to each well. Because we normalized everything against untreated cells, which were proliferating, our limit of detection would be close to 1 in 10^8 , so the results in Figure 1b are well within that range.

4. Figure 2: The different panels cited in the text do not match with the Figure, both for in vitro and in vivo experiments.

Corrected.

REVIEWERS' COMMENTS

Reviewer #1 (Remarks to the Author):

The authors have thoroughly answered all raised issues.

Reviewer #2 (Remarks to the Author):

Comments:

Most of the issues of my review have been addressed by the authors:

Answer 1. Ok

Answer 2. Ok. In order to be clearer, can the authors specify the concentration in term of "MIC value" all along the main text.

Answer 3. Ok. Please specify in the result section that in the *C. glabrata* sepsis model infection is not associated with animal death during the experiment period.

Answer 4. Ok.

Answer Fig 1c. Ok.

Answer Fig1: Ok.

Answer Fig 1g. Ok.

Answer Fig 2. Ok.

Answer Fig 4. Ok.

Answer statistical chapter. Ok.

Reviewer #3 (Remarks to the Author):

The authors have responded to all my queries and modified the manuscript accordingly. I have no further comments.

Our responses to Reviewer 2 are in blue font:

In order to be clearer, can the authors specify the concentration in term of "MIC value" all along the main text.

Done.

Please specify in the result section that in the *C. glabrata* sepsis model infection is not associated with animal death during the experiment period.

Done (lines 172-173).